# Coupled Variational Reinforcement Learning for Language Model General Reasoning

**Xueru Wen** [1 2]  **Jie Lou** [3]  **Yanjiang Liu** [1 2]  **Hongyu Lin** [1 2]  **Ben He** [1]
**Xianpei Han** [1 2]  **Le Sun** [1 2]  **Yaojie Lu** [1 2]  **Debing Zhang** [3]

## Abstract

While reinforcement learning has achieved impressive progress in language model reasoning, it is constrained by the requirement for verifiable rewards. Recent verifier-free RL methods address this limitation by utilizing the probabilities that LLMs generate reference answers as reward signals. However, these approaches typically sample reasoning traces conditioned only on the question. This design decouples reasoning-trace sampling from answer information, leading to inefficient exploration and incoherence between traces and final answers. In this paper, we propose *Coupled Variational Reinforcement Learning* (CoVRL), which bridges variational inference and reinforcement learning by coupling prior and posterior distributions through a hybrid sampling strategy. By constructing and optimizing a composite distribution that integrates these two distributions, CoVRL enables efficient exploration while preserving strong thought-answer coherence. Extensive experiments on mathematical and general reasoning benchmarks show that CoVRL improves performance by 12.4% over the base model and achieves an additional 2.3% improvement over state-of-the-art verifier-free RL baselines, providing a principled framework for enhancing the general reasoning capabilities of language models.

## 1. Introduction

Recent works (DeepSeek-AI et al., 2025; Yue et al., 2025; Yu et al., 2025a) explore reinforcement learning with verifi-

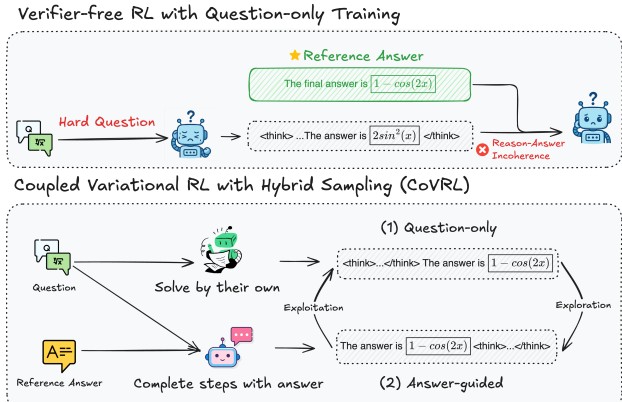

*Figure 1.* Comparison between verifier-free RL with question-only training and CoVRL. Unlike prior methods that sample reasoning traces conditioned only on the question, CoVRL couples question-conditioned prior sampling with answer-conditioned posterior sampling via a hybrid variational framework, enabling efficient exploration while preserving strong trace-answer coherence.

able rewards (RLVR) to enhance LLMs' reasoning capabilities, demonstrating impressive results in mathematical tasks. In this paradigm, the LLM generates a chain of thought (Wei et al., 2023) followed by an answer, and a rule-based program evaluates the final answer, assigning binary rewards based on correctness. The model is then optimized using policy gradient methods such as GRPO (Shao et al., 2024). However, these techniques require verifiable rewards, which restricts their applicability to other domains where formal verification of model answers is challenging. For instance, in chemistry, robust rule-based verification engines are rarely available for assessing semantic correctness across diverse formats of reaction equations.

A natural solution to this constraint is to introduce an LLM as a verifier (Ma et al., 2025), which performs a role similar to that of the reward model in reinforcement learning from human feedback (Ouyang et al., 2022). However, this approach requires a strong verifier and faces the risk of reward hacking (Skalse et al., 2025). To bypass the need for external verification modules, verifier-free methods (Yu et al., 2025b; Tang et al., 2025; Zhou et al., 2025a; Chen et al.,

[1]University of Chinese Academy of Sciences [2]Chinese Information Processing Laboratory, Institute of Software, Chinese Academy of Sciences [3]Xiaohongshu Inc. Correspondence to: Yaojie Lu <luyaojie@iscas.ac.cn>, Jie Lou <loujie0822@gmail.com>.

*Proceedings of the 43rd International Conference on Machine Learning*, Seoul, South Korea. PMLR 306, 2026. Copyright 2026 by the author(s).

2024) have emerged that utilize the probabilities LLMs assign to correct answers as reward signals. These methods view answer generation as a process that relies on a set of implicit reasoning steps. In this formulation, each potential reasoning trace is treated as a latent variable contributing to the final answer. The probability of the correct answer is obtained by marginalizing over all plausible reasoning traces conditioned on the input question. In practice, the model is trained using reinforcement learning by sampling different intermediate thoughts for each question and using the probability of generating the correct answer from each thought as the reward.

While these methods eliminate external verifiers, they typically rely on question-only generation, i.e., the model observes the question and then attempts to generate reasoning steps leading to the final answer. As shown in Figure 1, this strategy faces two fundamental challenges: **(1) Low sample efficiency**, particularly for difficult questions where the model struggles to produce useful reasoning traces without guidance; and **(2) Potential incoherence** between reasoning traces and final answers due to the lack of answer guidance during trace generation, where even correct reasoning may receive low rewards because the final answer is expressed in a different format from the ground truth.

In this work, we propose *Coupled Variational Reinforcement Learning* (CoVRL), which frames reasoning training as a variational optimization problem that couples prior and posterior distributions. The core idea is to address the challenges of question-only training by establishing coupling between two complementary distributions for sampling reasoning traces during training. We leverage both prior and posterior distributions as variational components, corresponding to two complementary generation modes: **(1) question-only generation**, which samples traces from the prior distribution, reflecting real inference conditions; and **(2) answer-guided generation**, which samples traces from the posterior distribution to generate coherent reasoning that leads to the correct answer. This dual-mode strategy provides answer guidance during training while ensuring that the learned reasoning patterns transfer effectively to the inference phase, improving sample efficiency and mitigating incoherence between reasoning traces and answers.

To implement this, we construct a composite distribution that combines the prior and posterior probabilities, establishing a coupled training framework. Since directly sampling from this composite distribution is computationally complex, we adopt a hybrid sampling strategy where we randomly select between the two generation modes for each training example. We then maximize a variational lower bound, which includes a reconstruction term for answer prediction and a regularization term to ensure transferability to inference. Through importance weighting, we enable seamless training across both modes using the same underlying language model with different prompt templates.

In summary, our main contributions include:

- We formulate reasoning optimization as a variational inference problem and introduce a composite distribution that theoretically unifies prior and posterior generation modes within a tractable framework.

- We propose a hybrid sampling strategy that balances answer-guided learning and inference-time transferability, thus increasing sampling efficiency and addressing trace-answer coherence challenges.

- We demonstrate consistent improvements across diverse benchmarks, achieving a 12.4% improvement over the base model and a 2.3% improvement over the strongest baseline[1].

## 2. Methodology

In this section, we introduce Coupled Variational Reinforcement Learning. Figure 2 illustrates the overall framework.

### 2.1. Preliminaries

**Reinforcement Learning with Verifiable Rewards** Recent work (Zhao et al., 2025; Yu et al., 2025a) has explored using reinforcement learning to optimize language models for reasoning tasks. Given a reward function $R(x, y)$ that evaluates the correctness of the generated reasoning and answer, the objective is:

$$\max_{\theta} \mathbb{E}_{x \sim \mathcal{D}, y \sim p_{\theta}(\cdot|x)}[R(x, y)] \qquad (1)$$

where $p_{\theta}$ denotes the model distribution parameterized by $\theta$, $x$ is the question, and $y$ represents the generated reasoning and answer. This objective can be optimized using policy gradient methods like GRPO (Shao et al., 2024). However, these approaches require verifiable reward signals, limiting their applicability to domains without accessible verifiers.

**Variational Inference with Latent Variables** Variational inference (Li et al., 2019; Kingma & Welling, 2022) provides a principled framework for learning models with latent variables. Consider a generative process where observed data $y$ depends on latent variables $z$:

$$p(y) = \int p(y|z)p(z)dz \qquad (2)$$

Since this marginal likelihood is typically intractable, we introduce a variational posterior $q_{\phi}(z|y)$. By applying

---

[1]The code and data associated with this work will be available at `https://github.com/wenxueru/CoVRL`.

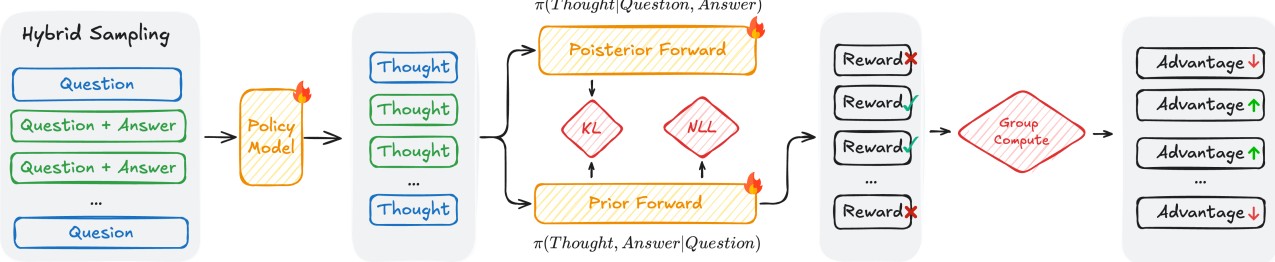

*Figure 2.* CoVRL employs hybrid sampling between prior $p_\phi(z|x)$ and posterior $q_\psi(z|x, y)$ to generate reasoning traces. It optimizes the reconstruction term using GRPO and NLL loss, with KL regularization applied to ensure training-inference coherence.

Jensen's inequality, we obtain:

$$
\begin{aligned}
\log p(y) &= \log \mathbb{E}_{q_\phi(z|y)} \left[ \frac{p(y|z)p(z)}{q_\phi(z|y)} \right] \\
&\geq \mathbb{E}_{q_\phi(z|y)}[\log p_\theta(y|z)] - D_{\mathrm{KL}}(q_\phi(z|y)\|p(z))
\end{aligned}
\tag{3}
$$

Here, $p_\theta(y|z)$ is the decoder distribution that reconstructs the observed data given the latent variables, while $q_\phi(z|y)$ is the encoder distribution that infers latent representations from the observed data. This lower bound is called the evidence lower bound (ELBO), which is widely used in variational autoencoders (Kingma & Welling, 2022) and diffusion models (Ho et al., 2020).

## 2.2. Coupled Variational Reinforcement Learning

In reasoning tasks, we treat reasoning traces as latent variables $z$ that mediate between questions $x$ and answers $y$, thus reformulating Equation 2 as:

$$
p(y|x) = \int p(y|z, x)p(z|x)dz
\tag{4}
$$

Previous approaches generally optimize this objective by directly sampling from the prior distribution $p_\phi(z|x)$. However, these approaches face two main challenges: **(1) Low sampling efficiency**: reasoning traces sampled from the prior $p_\phi(z|x)$ may fail to effectively predict the target answer when the question is too difficult for the model, leading to inefficient exploration of the reasoning space. **(2) Trace-answer incoherence**: without access to the target answer, even correct reasoning traces can receive low rewards due to mismatched answer formulations or representations compared to the ground-truth answer $y$.

In response to these limitations, we introduce a variational distribution $q(z)$. Following the standard variational infer-

ence framework, we derive the following objective:

$$
\begin{aligned}
\log p_\theta(y|x) &= \log \int p_\theta(y|z, x)p_\phi(z|x)dz \\
&= \log \int q(z)\frac{p_\theta(y|z, x)p_\phi(z|x)}{q(z)}dz \\
&= \log \mathbb{E}_{q(z)} \left[ \frac{p_\theta(y|z, x)p_\phi(z|x)}{q(z)} \right] \\
&\geq \mathbb{E}_{q(z)} \left[ \log \frac{p_\theta(y|z, x)p_\phi(z|x)}{q(z)} \right] \text{ (Jensen's inequality)} \\
&= \mathbb{E}_{q(z)}[\log p_\theta(y|z, x)] - D_{\mathrm{KL}}(q(z)\|p_\phi(z|x))
\end{aligned}
\tag{5}
$$

This ELBO consists of two components: a reconstruction term $\mathbb{E}_{q(z)}[\log p_\theta(y|z, x)]$ that encourages generating reasoning traces leading to the correct answer, and a KL regularization term $D_{\mathrm{KL}}(q(z)\|p_\phi(z|x))$ that constrains deviation from the prior distribution.

Previous approaches can be viewed as setting $q(z) = p_\phi(z|x)$, which suffers from the aforementioned limitations. A natural alternative is to use the posterior distribution as $q(z)$, as in VAE (Kingma & Welling, 2022), which samples reasoning traces conditioned on both the question and the target answer, i.e., $q(z) = q_\psi(z|x, y)$. This enables generating more relevant reasoning paths and improves sampling efficiency compared to prior-only sampling.

However, this introduces a fundamental training-inference mismatch. During training, we optimize the posterior distribution with access to target answers, but at inference time, we must rely on the prior distribution $p_\phi(z|x)$, which is conditioned only on the question. Even with KL divergence regularization $D_{\mathrm{KL}}(q_\psi(z|x, y)\|p_\phi(z|x))$ that encourages the posterior to stay close to the prior, there is no guarantee that optimizing the posterior will improve the prior due to the asymmetric nature of reverse KL divergence (Chan et al., 2022). Specifically, while $D_{\mathrm{KL}}(q_\psi\|p_\phi)$ constrains the posterior to avoid low-probability regions of the prior, it cannot ensure that the posterior covers all high-probability regions of the prior. This results in inadequate training for some regions of the prior, potentially leading to poor inference

performance when sampling from these regions. In response to the difficulties of sampling independently from the prior $p_\phi(z|x)$ and posterior $q_\psi(z|x, y)$ distributions, we introduce CoVRL, which establishes coupling between these distributions through complementary mechanisms.

## 2.3. Composite Distribution

Rather than relying on either distribution independently, we propose to couple them through a unified framework. We first construct a composite distribution that mixes the prior and posterior at each token position. For a reasoning trace $z = (z_1, \ldots, z_T)$, the **composite distribution** is defined through token-level averaging:

$$p'(z_t|z_{<t}, x, y) = \frac{1}{2}p_\phi(z_t|z_{<t}, x) + \frac{1}{2}q_\psi(z_t|z_{<t}, x, y) \quad (6)$$

where equal weighting encourages balanced gradient contributions from both the prior and the posterior during optimization. The sequence-level probability is obtained by the standard autoregressive factorization:

$$p'(z|x, y) = \prod_{t=1}^{T} p'(z_t|z_{<t}, x, y) \quad (7)$$

This composite distribution couples the prior $p_\phi(z|x)$ and posterior $q_\psi(z|x, y)$. By including both distributions in $p'$, this construction enables direct optimization of the prior through answer-guided signals while ensuring the posterior aligns with inference conditions, avoiding the coverage issues of posterior-only training.

To derive the optimization objective for this composite distribution, we replace $q(z)$ with $p'(z|x, y)$ in Equation 5:

$$\log p(y|x) = \log \int p(y|z, x)p(z|x)dz$$
$$= \log \mathbb{E}_{p'(z|x,y)} \left[ \frac{p(y|z, x)p(z|x)}{p'(z|x, y)} \right]$$
$$\geq \underbrace{\mathbb{E}_{p'(z|x,y)}[\log p_\theta(y|z, x)]}_{\text{Reconstruction term}} - \underbrace{D_{\text{KL}}(p'(z|x, y)\|p_\phi(z|x))}_{\text{Regularization term}}$$
$$(8)$$

This ELBO consists of two components:

**Reconstruction term** $\mathbb{E}_{p'(z|x,y)}[\log p_\theta(y|z, x)]$ measures the model's ability to predict the correct answer $y$ given the reasoning trace $z$ and question $x$, serving as the primary target for improving reasoning quality.

**Regularization term** $D_{\text{KL}}(p'(z|x, y)\|p_\phi(z|x))$ constrains the composite distribution to remain close to the prior distribution, ensuring that the learned reasoning patterns transfer effectively to inference.

In practice, all three components $p_\phi(z|x)$, $q_\psi(z|x, y)$, and $p_\theta(y|z, x)$ can be implemented using a single LLM with

different prompts: $p_\phi(z|x)$ conditions on the input question, $q_\psi(z|x, y)$ conditions on both question and answer, and $p_\theta(y|z, x)$ predicts the answer given the reasoning trace. Thus, the subscripts $\phi, \psi, \theta$ denote different conditioning contexts rather than separate model parameters.

## 2.4. Hybrid Sampling and Coupled Optimization

To optimize the composite distribution, a straightforward approach would be to directly sample from $p'(z|x, y)$ during training. However, this would require real-time mixing of prior and posterior distributions for generating each new token, demanding additional computation and complex modifications to prevalent LLM-RL frameworks (Sheng et al., 2024; Hu et al., 2024), which typically utilize efficient inference engines such as SGLang (Team, 2024) and vLLM (Kwon et al., 2023) for rollout.

Therefore, instead of sampling from the actual composite distribution, we adopt off-policy **hybrid sampling**. For each training sample, we randomly sample from the prior with probability $\alpha$ or from the posterior with probability $1 - \alpha$:

$$p_{\text{hybrid}}(z|x, y) = \begin{cases} p_\phi(z|x) & \text{with probability } \alpha \\ q_\psi(z|x, y) & \text{with probability } 1 - \alpha \end{cases}$$
$$(9)$$

This approach ensures that both distributions are sampled, enabling joint optimization of exploratory and answer-guided reasoning. Although $p_{\text{hybrid}}$ differs from the target composite distribution $p'$, we show below that we can still optimize $p'$ by appropriately reweighting samples through importance sampling.

**For the composite distribution** $p'(z|x, y)$ defined through token-level mixing, the gradient becomes:

$$\nabla_{\phi,\psi}\mathbb{E}_{p'(z|x,y)}[\log p_\theta(y|z, x)] =$$
$$\mathbb{E}_{p'(z|x,y)}[\log p_\theta(y|z, x)\nabla_{\phi,\psi}\log p'(z|x, y)] \quad (10)$$

Optimizing the composite distribution requires policy gradient methods due to the discrete sampling process. We employ Group Relative Policy Optimization (GRPO) (Shao et al., 2024) to optimize the reconstruction term with the following objective:

$$L_{\phi,\psi} = \mathbb{E}_{p'(z|x,y)}\left[\sum_{t=1}^{T}\min(r_t\hat{A}, \text{clip}(r_t, 1 - \epsilon, 1 + \epsilon)\hat{A})\right]$$
$$(11)$$

where $r_t = \frac{p'_{\text{new}}(z_t|z_{<t}, x, y)}{p'_{\text{old}}(z_t|z_{<t}, x, y)}$ represents the token-level probability ratio between the new and old policies, and $\epsilon$ is the clipping parameter. The advantage $\hat{A} = \log p_{\theta_{\text{old}}}(y|z, x) - \bar{R}$ is computed using group relative estimation, where $\bar{R}$ is the average reward across the batch. Importantly, the reward is computed using the model's own answer prediction probability, eliminating the need for external verifiers.

| Prior Distribution Template |
|---|
| <\|im_start\|>system
A conversation between User and Assistant. The user asks a question, and the Assistant solves it. The assistant first thinks about the reasoning process in the mind and then provides the user with the answer. The reasoning process and answer are enclosed within <think> </think> and <answer> </answer> tags, respectively, i.e., <think>reasoning process here</think><answer>answer here</answer>.
<\|im_end\|>

<\|im_start\|>user
{question}
<\|im_end\|>

<\|im_start\|>assistant
<think>
{thought}
</think>
<answer>
{answer}
</answer>
<\|im_end\|> |

| Posterior Distribution Template |
|---|
| <\|im_start\|>system
A conversation between User and Assistant. The user asks a question, and the Assistant solves it. The assistant provides the final answer first, then follows up with a comprehensive reasoning process. The answer and reasoning process are enclosed within <answer> </answer> and <think> </think> tags, respectively, i.e., <answer>answer here</answer><think>reasoning process here</think>.
<\|im_end\|>

<\|im_start\|>user
{question}
<\|im_end\|>

<\|im_start\|>assistant
<answer>
{answer}
</answer>
<think>
{thought}
</think>
<\|im_end\|> |

*Figure 3.* Prompt templates after applying chat template for Prior and Posterior distributions. The key difference lies in the order of reasoning and answer components within the assistant's response.

To account for the distribution mismatch between our hybrid sampling strategy and the target composite distribution, we optimize the composite distribution through importance sampling. Specifically, we replace $p'_{\text{old}}(z_t|z_{<t}, x, y)$ in Equation 11 with the actual sampling distribution $p_{\text{hybrid}}(z_t|z_{<t}, x, y)$ to enable mathematically principled optimization of the target composite distribution. Thus, the token-level importance ratio $r_t$ becomes:

$$r_t = \frac{p'_{\text{new}}(z_t|z_{<t}, x, y)}{p_{\text{hybrid}}(z_t|z_{<t}, x, y)} \qquad (12)$$

where $p_{\text{hybrid}}(z_t|z_{<t}, x, y)$ equals $p_\phi(z_t|z_{<t}, x)$ for samples drawn from the prior distribution, and $q_\psi(z_t|z_{<t}, x, y)$ for samples drawn from the posterior distribution. In practice, this importance ratio can be computed via two forward passes for each sampled reasoning trace $z$: one with the prior template to obtain $p_\phi(z|x)$ and one with the posterior template to obtain $q_\psi(z|x, y)$.

**For the prior distribution** $p_\phi(z|x)$, the gradient is:

$$\nabla_\phi \mathbb{E}_{p_\phi(z|x)}[\log p_\theta(y|z, x)] = \\ \mathbb{E}_{p_\phi(z|x)}[\log p_\theta(y|z, x)\nabla_\phi \log p_\phi(z|x)] \qquad (13)$$

An important observation is that optimizing the prior distribution alone corresponds to standard maximum likelihood estimation, which can be efficiently computed using negative log-likelihood loss. However, applying NLL loss to all reasoning traces is problematic, as some traces may be of low quality. Since our model also serves as a reward model that evaluates reasoning quality through $\log p_\theta(y|z, x)$, we selectively reinforce high-quality traces. Therefore, we only compute the loss on samples that produce valid formatted outputs and have positive advantage $\hat{A} > 0$.

**2.5. Optimizing KL Divergence**

There are two approaches for optimizing the KL divergence term: (1) incorporating KL as part of the reward function as in PPO (Schulman et al., 2017), or (2) treating KL as a separate loss term as in GRPO (Shao et al., 2024). We adopt the latter approach, which we find is more stable.

Optimizing KL divergence as a loss function requires a non-negative and low-variance KL estimator. To estimate $D_{\text{KL}}(p'(z|x, y)\|p_\phi(z|x))$ in our composite distribution setting, we extend the low-variance estimators of Schulman (2020). Specifically, we derive different forms depending on

the sampling distribution using importance sampling with Bregman divergence-based control variates.

**When sampling from the prior distribution:**

$$D_{\mathrm{KL}}^{\mathrm{prior}} = \frac{p'(z|x,y)}{p_\phi(z|x)} \log \frac{p'(z|x,y)}{p_\phi(z|x)} - \frac{p'(z|x,y)}{p_\phi(z|x)} + 1 \quad (14)$$

**When sampling from the posterior distribution:**

$$D_{\mathrm{KL}}^{\mathrm{posterior}} = \frac{p'(z|x,y)}{q_\psi(z|x,y)} \log \frac{p'(z|x,y)}{p_\phi(z|x)} + \frac{p'(z|x,y)}{q_\psi(z|x,y)} - 1 \quad (15)$$

The key difference between these estimators lies in the sign of the Bregman correction term $\left(\frac{p'(z|x,y)}{p_{\mathrm{hybrid}}(z|x,y)} - 1\right)$. This correction serves as a control variate that reduces estimation variance without introducing bias, leveraging the fact that:

$$\mathbb{E}_{p_{\mathrm{hybrid}}}\left[\frac{p'(z|x,y)}{p_{\mathrm{hybrid}}(z|x,y)} - 1\right] = 0 \quad (16)$$

The different signs ensure that the estimator remains non-negative regardless of whether the trace is sampled from the prior or posterior distribution. Moreover, we apply soft clipping to the importance sampling ratios in both estimators when they exceed a threshold to prevent numerical overflow from the exponential terms while still providing stable gradients. In cases where the model generation reaches the maximum response length, we skip computing the KL loss for these samples. Appendix A provides a more intuitive discussion of these KL estimators.

## 2.6. Prompt Template and Tokenization

We follow the template format suggested in previous work (DeepSeek-AI et al., 2025), with simple extensions to suit our hybrid sampling strategy. The prompt templates for prior and posterior generation are shown in Figure 3. For tokenization, we separately encode special tokens including `<think>`, `</think>`, `<answer>` and `</answer>` to ensure consistent token ID sequences. However, during model generation, these special tokens may still be merged with adjacent tokens. Therefore, we use strings as stop sequences during model generation. After generation is completed, we decode the outputs and re-encode them with separately encoded special tokens.

## 3. Experiments

### 3.1. Setup

**Dataset.** Following previous work, we use the dataset curated by (Ma et al., 2025), which was sourced from WebInstruct (Yue et al., 2024). We retain only the non-mathematical questions to assess training improvements on general reasoning abilities. Beyond this selection, we apply

no additional filtering to evaluate the algorithm's robustness across diverse question types, difficulty levels, and quality variations.

**Implementation.** We conduct experiments primarily using Qwen2.5-7B-Base (Qwen et al., 2025). We directly fine-tune the base model without an intermediate supervised fine-tuning stage. We implement our RL training pipeline using the verl framework (Sheng et al., 2024). During training, we use a batch size of 192 questions and generate 8 responses per question during rollout. For rollout sampling, we use temperature = 1.0, top_p = 1.0, and max_tokens = 2048 to ensure diverse response generation while maintaining reasonable response lengths. The clip threshold in the policy gradient loss is set to 0.3 given the off-policy nature of our algorithm. We use a cosine learning rate scheduler with 64 warmup steps and a peak learning rate of 1e-6.

**Baselines.** Our baseline selection focuses on other verifier-free methods that do not require external verification models during training: VeriFree (Zhou et al., 2025a), RLPR (Yu et al., 2025b), JLB (Tang et al., 2025), RAVR (Lin et al., 2025), and LaTRO (Chen et al., 2024). We summarize the gradient estimators of these methods in Appendix B. To eliminate the influence of different RL techniques and focus on the gradient estimator, we train all baselines on the Qwen2.5-7B-base model using GRPO (Shao et al., 2024) with identical hyperparameters on the filtered dataset.

**Evaluation.** We evaluate our method on a comprehensive suite of reasoning benchmarks. For general reasoning, we use MMLU-Pro (Wang et al., 2024), GPQA (Rein et al., 2023), and TheoremQA (Chen et al., 2023). For mathematical reasoning, we use AIME'24 (Mathematical Association of America, 2024), AQuA (Ling et al., 2017), CARP-EN (Zhang et al., 2023), MATH-500 (Lightman et al., 2023), Minerva (Lewkowycz et al., 2022), and SAT-Math (mcaleste, 2023). For evaluation, we use temperature=0.6 and max_tokens=4096. We utilize Math-Verify to check answer correctness; more details are provided in Appendix C. To reduce evaluation variance, we repeat the evaluation $N$ times for each dataset, where $N$ depends on the test set size. We report the average performance across these $N$ runs as Average@$N$ to account for sampling variance.

### 3.2. Main Results

Table 1 presents our main experimental results across all benchmarks. Our approach achieves consistent improvements over the base model while outperforming the strongest baseline by 2.3% on average, demonstrating the effectiveness of our coupled optimization framework.

**CoVRL achieves consistent performance improvements across benchmarks.** CoVRL demonstrates substantial improvements over the base model across all tasks, achieving

*Table 1.* Performance on benchmarks. Results are reported as Average@N where N indicates the number of independent evaluation runs to account for sampling variance. The Overall column presents the arithmetic mean across all benchmark tasks.

| Method | General Reasoning | | | Mathematical Reasoning | | | | | | Overall |
|---|---|---|---|---|---|---|---|---|---|---|
| | GPQA | MMLU-Pro | TheoremQA | AIME'24 | AQuA | CARP-EN | MATH-500 | Minerva | SAT-Math | |
| | Avg@4 | Avg@1 | Avg@2 | Avg@32 | Avg@4 | Avg@2 | Avg@4 | Avg@4 | Avg@32 | |
| Base Model | 26.1 | 36.7 | 25.2 | 2.7 | 55.6 | 54.3 | 44.7 | 18.6 | 76.5 | 37.8 |
| VeriFree | 28.9 | 44.1 | 33.4 | 5.0 | 72.6 | 62.8 | 59.5 | 24.0 | 93.3 | 47.1 |
| JLB | **31.6** | 42.7 | 31.9 | 4.8 | 69.5 | 63.4 | 57.6 | 23.6 | 93.7 | 46.5 |
| LaTRO | 31.0 | 42.7 | 32.8 | 4.0 | 67.6 | 50.5 | 59.3 | 24.4 | 90.1 | 44.7 |
| RAVR | 30.2 | 44.5 | 34.8 | 6.3 | 72.4 | 63.2 | 61.2 | 23.3 | 94.1 | 47.8 |
| RLPR | 31.3 | 44.9 | 33.5 | 6.5 | 72.3 | 62.9 | 61.2 | 24.7 | 93.8 | 47.9 |
| **CoVRL (Ours)** | 30.4 | **46.5** | **36.3** | **7.5** | **77.3** | **65.1** | **66.3** | **25.5** | **97.1** | **50.2** |

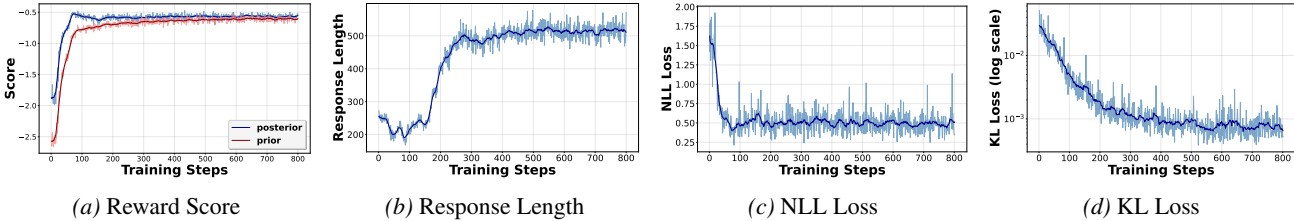

*(a)* Reward Score        *(b)* Response Length        *(c)* NLL Loss        *(d)* KL Loss

*Figure 4.* Training dynamics of CoVRL across different metrics. We observe stable improvements in reasoning quality alongside effective optimization of both reconstruction and regularization objectives.

the highest overall performance at 50.2%, representing a 12.4% improvement. In comparison with other baselines, CoVRL achieves the strongest overall gains. These results suggest that coupling prior and posterior sampling provides more effective training signals.

**Reasoning capabilities learned by CoVRL are generalizable to mathematical reasoning.** Despite training on non-mathematical questions, our approach shows substantial gains on mathematical benchmarks. This validates that general reasoning capabilities developed through diverse problem-solving can transfer effectively, highlighting the value of general reasoning skill development.

### 3.3. Training Dynamics

We present the evolution of key metrics during training in Figure 4 to better understand the behavior of the algorithm. These metrics characterize different aspects of CoVRL, including reward improvement, response-length evolution, answer reconstruction, and distributional regularization. This leads to several observations:

**The posterior distribution is effective in providing guidance.** Figure 4a shows steady reward score improvements throughout training, with the posterior distribution consistently outperforming the prior distribution. This persistent score gap validates our answer-guided sampling strategy and confirms that posterior sampling enables more efficient exploration of high-quality reasoning paths. Meanwhile, the

prior reward also improves steadily, suggesting that the guidance provided by the posterior can be effectively transferred to inference-time generation.

**CoVRL enhances reasoning capabilities with prolonged chain-of-thought traces.** Figure 4b shows a stable increase in response length, indicating that the model progressively generates more detailed reasoning processes. This trend suggests that CoVRL successfully encourages elaborate chain-of-thought reasoning with more thorough step-by-step explanations.

**Regularization provides stable optimization dynamics.** Figures 4c and 4d show stable downward trends in NLL and KL losses. The decreasing losses indicate improved answer prediction and successful regularization, confirming that our variational objective effectively balances the reconstruction and regularization terms. Overall, these training dynamics demonstrate that CoVRL improves reasoning quality while maintaining stable optimization throughout training.

### 3.4. Hybrid Sampling Strategy

We examine the impact of our hybrid sampling strategy by varying the mixing ratio $\alpha$ between prior $p_\phi(z|x)$ and posterior $q_\psi(z|x, y)$ distributions. Here $\alpha$ represents the probability of sampling from the prior distribution. By default, we set $\alpha = 0.5$. Figure 5 demonstrates that low prior sampling probability ($\alpha = 0.1$) outperforms high prior sampling probability ($\alpha = 0.9$). This highlights the

*Table 2.* Performance comparison across different base models of various model architectures and sizes.

| Model | General | | | Mathematical | | | | | | |
| | GPQA Avg@4 | MMLU-Pro Avg@1 | TheoremQA Avg@2 | AIME'24 Avg@16 | AQuA Avg@4 | CARP-EN Avg@2 | MATH-500 Avg@4 | Minerva Avg@4 | SAT-Math Avg@32 | Overall |
|---|---|---|---|---|---|---|---|---|---|---|
| Qwen2.5-7B | 26.1 | 36.7 | 25.2 | 2.7 | 55.6 | 54.3 | 44.7 | 18.6 | 76.5 | 37.8 |
| + CoVRL | $30.4_{+4.3}$ | $46.5_{+9.8}$ | $36.3_{+11.1}$ | $7.5_{+4.8}$ | $77.3_{+21.7}$ | $65.1_{+10.8}$ | $66.3_{+21.6}$ | $25.5_{+6.9}$ | $97.1_{+20.6}$ | $50.2_{+12.4}$ |
| Qwen2.5-14B | 28.0 | 38.5 | 25.4 | 3.0 | 63.3 | 57.3 | 44.8 | 22.1 | 82.5 | 40.5 |
| + CoVRL | $36.6_{+8.6}$ | $57.0_{+18.5}$ | $42.4_{+17.0}$ | $7.9_{+4.9}$ | $81.8_{+18.5}$ | $64.0_{+6.7}$ | $71.5_{+26.7}$ | $33.7_{+11.6}$ | $95.8_{+13.3}$ | $54.5_{+14.0}$ |
| Qwen3-8B | 36.9 | 51.0 | 32.8 | 4.0 | 68.6 | 59.4 | 53.6 | 30.0 | 90.5 | 47.4 |
| + CoVRL | $37.6_{+0.7}$ | $59.6_{+8.6}$ | $43.7_{+10.9}$ | $9.2_{+5.2}$ | $80.5_{+11.9}$ | $66.1_{+6.7}$ | $74.5_{+20.9}$ | $35.2_{+5.2}$ | $97.6_{+7.1}$ | $56.0_{+8.6}$ |
| Qwen3-14B | 37.5 | 57.6 | 37.6 | 7.8 | 75.3 | 64.6 | 67.2 | 33.5 | 93.1 | 52.7 |
| + CoVRL | $42.7_{+5.2}$ | $63.1_{+5.5}$ | $46.3_{+8.7}$ | $9.5_{+1.7}$ | $83.9_{+8.6}$ | $65.9_{+1.3}$ | $76.1_{+8.9}$ | $38.1_{+4.6}$ | $97.4_{+4.3}$ | $58.1_{+5.4}$ |

*Table 3.* Performance comparison across different training data compositions.

| Training Data | General | | | Mathematical | | | | | | |
| | GPQA Avg@4 | MMLU-Pro Avg@1 | TheoremQA Avg@2 | AIME'24 Avg@16 | AQuA Avg@4 | CARP-EN Avg@2 | MATH-500 Avg@4 | Minerva Avg@4 | SAT-Math Avg@32 | Overall |
|---|---|---|---|---|---|---|---|---|---|---|
| Base Model | 26.1 | 36.7 | 25.2 | 2.7 | 55.6 | 54.3 | 44.7 | 18.6 | 76.5 | 37.8 |
| Non-Math Only | $30.4_{+4.3}$ | $46.5_{+9.8}$ | $36.3_{+11.1}$ | $7.5_{+4.8}$ | $77.3_{+21.7}$ | $65.1_{+10.8}$ | $66.3_{+21.6}$ | $25.5_{+6.9}$ | $97.1_{+20.6}$ | $50.2_{+12.4}$ |
| Math Only | $28.9_{+2.8}$ | $42.7_{+6.0}$ | $31.8_{+6.6}$ | $4.2_{+1.5}$ | $72.1_{+16.5}$ | $53.1_{-1.2}$ | $60.1_{+15.4}$ | $20.6_{+2.0}$ | $94.3_{+17.8}$ | $45.3_{+7.5}$ |

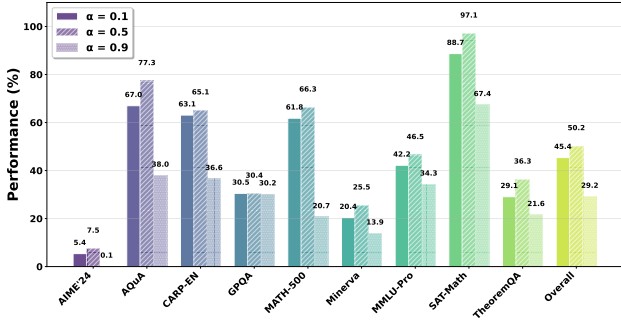

*Figure 5.* Impact of hybrid sampling probability $\alpha$.

important role of the posterior distribution in our algorithm. When $\alpha = 0.9$, the model primarily samples from the prior distribution, and we observe reduced reasoning chain length. We attribute this to the difficulty of improving rewards, which leads the model to prioritize minimizing the KL loss and generating shorter sequences. When $\alpha = 0.1$, reasoning chain length increases, and posterior-dominated sampling achieves better performance than prior-dominated sampling. However, due to training-inference mismatch, performance remains inferior to balanced sampling. Training dynamics under these settings are provided in Appendix D.

### 3.5. Training Model and Data

**CoVRL is robust across different base models.** We evaluate CoVRL on the Qwen2.5-base and Qwen3-base models, ranging from 7B to 14B parameters. Note that for the Qwen3-base experiments, we replace the `<think>` and `</think>` tags with `<thinking>` and `</thinking>`,

respectively, because Qwen3-base does not reliably utilize the `<think>` and `</think>` tags in prompts but responds well to similar alternatives. As shown in Table 2, CoVRL delivers consistent performance improvements across all tested models, with gains ranging from 8.6% to 14.0% across different model sizes, demonstrating its robustness and effective scaling with model capacity.

**CoVRL learns generalizable reasoning abilities from training data.** We also evaluate CoVRL with different training data compositions. The results in Table 3 reveal that training on both math-only and non-math-only data significantly improves base model performance. Notably, models trained solely on mathematical data demonstrate enhanced performance on non-mathematical reasoning tasks, and similarly, models trained on non-mathematical data improve on mathematical tasks. This indicates that our method enables models to acquire generalizable reasoning capabilities that transfer across different domains, suggesting that CoVRL develops fundamental reasoning patterns rather than domain-specific solutions.

## 4. Related Work

**Verifier-free Reinforcement Learning.** While RLVR (Cui et al., 2025; Yu et al., 2025a; Yang et al., 2025; DeepSeek-AI et al., 2025) has emerged as common practice, these approaches require robust verifiers, limiting their applicability to broader reasoning tasks. Recent work such as LaTRO (Chen et al., 2024) eliminates this dependency by formulating reasoning as sampling from latent distributions. Sub-

sequent methods such as JLB (Tang et al., 2025), VeriFree (Zhou et al., 2025a), and RLPR (Yu et al., 2025b) extend this framework using probability-based rewards, but these prior-based sampling methods suffer from limited sample efficiency and reasoning-answer misalignment. RAVR (Lin et al., 2025) instead utilizes posterior distributions for sampling. Our approach establishes explicit coupling between prior and posterior through a composite distribution and hybrid sampling strategy.

**Self-improving Language Models.** Recent work enhances language models by leveraging self-generated training signals, including preference learning (Yuan et al., 2025), iterative DPO (Chen et al., 2025; Rafailov et al., 2024), and test-time voting (Zuo et al., 2025). Alternative approaches construct rewards from reference answers, such as the verifier-free methods discussed above and Zhou et al. (2025b), which introduce posterior distributions and update separate prior and posterior models using IWAE (Burda et al., 2016) with an EM algorithm. In comparison, we optimize a single composite distribution with verifier-free rewards through online RL training.

## 5. Conclusion

In this paper, we introduce Coupled Variational Reinforcement Learning for enhancing language model reasoning without external verifiers. Our method enables joint optimization of question-only and answer-guided generation through composite distributions. By establishing coupling via hybrid sampling, CoVRL balances efficient exploration with inference-time transferability.

## Acknowledgements

We sincerely thank the reviewers for their insightful comments and valuable suggestions. This work was supported by the Natural Science Foundation of China (No. 62536008, 62572456, 62272439, 62306303).

## Impact Statement

This paper presents work whose goal is to advance the field of Machine Learning. Our method improves language model reasoning capabilities without requiring external verifiers, potentially democratizing access to advanced AI reasoning. While this enables beneficial applications in education and scientific discovery, we acknowledge that enhanced reasoning abilities could be misused for generating sophisticated misinformation. We encourage the community to consider safety mechanisms and ethical guidelines when deploying such capabilities.

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

# A. KL Divergence Estimator

In this section, we provide the derivation and intuition for our KL divergence estimators used in CoVRL.

## A.1. Derivation via Control Variates and Bregman Divergence

Our KL estimators are derived following the control variate approach proposed by Schulman (2020). The key idea is to construct unbiased, low-variance estimators of KL divergence when we can only sample from one of the distributions.

**Reverse KL** $D_{\mathrm{KL}}(q\|p)$: When sampling from $q$ to estimate $D_{\mathrm{KL}}(q\|p) = \mathbb{E}_{x\sim q}[\log q(x) - \log p(x)]$, the naive unbiased estimator is $-\log r$ where $r = \frac{p(x)}{q(x)}$, but this has high variance as it can be negative even though KL is always non-negative. To reduce variance while maintaining unbiasedness, we employ a **control variate**: we add a zero-mean term that is negatively correlated with the naive estimator. Since $\mathbb{E}_{x\sim q}[r - 1] = 0$, we can construct the estimator:

$$-\log r + \lambda(r - 1) \tag{17}$$

which is unbiased for any $\lambda$. Setting $\lambda = 1$ yields:

$$(r - 1) - \log r \tag{18}$$

**Forward KL** $D_{\mathrm{KL}}(p\|q)$: When sampling from $q$ to estimate $D_{\mathrm{KL}}(p\|q) = \mathbb{E}_{x\sim p}[\log p(x) - \log q(x)]$, we need importance weighting. The naive importance-weighted estimator is $r \log r$ where $r = \frac{p(x)}{q(x)}$. Applying the control variate $r - 1$ (which has zero mean under $q$), we obtain:

$$r \log r - (r - 1) \tag{19}$$

Both forms are always non-negative because they measure the **Bregman divergence** between $\log(x)$ and its tangent line at $x = 1$. Since log is concave, we have $\log(x) \leq x - 1$, ensuring non-negativity. This generalizes to f-divergences via the formula $f(r) - f'(1)(r - 1)$, which measures the distance between a convex function $f$ and its tangent at $r = 1$.

## A.2. KL Estimators Under Hybrid Sampling

Recall that our composite distribution is defined through token-level mixing:

$$p'(z_t|z_{<t}, x, y) = \frac{1}{2}p_\phi(z_t|z_{<t}, x) + \frac{1}{2}q_\psi(z_t|z_{<t}, x, y) \tag{20}$$

Let $r_t = \frac{q_\psi(z_t|z_{<t}, x, y)}{p_\phi(z_t|z_{<t}, x)}$ denote the token-level likelihood ratio between posterior and prior distributions. Then:

$$p'(z_t|z_{<t}, x, y) = p_\phi(z_t|z_{<t}, x)\left(\frac{1}{2} + \frac{1}{2}r_t\right) \tag{21}$$

We need to estimate $D_{\mathrm{KL}}(p'(z_t|z_{<t}, x, y)\|p_\phi(z_t|z_{<t}, x))$ at each token position. In our hybrid sampling strategy, we sample complete sequences from either the prior $p_\phi(z|x)$ or posterior $q_\psi(z|x, y)$, rather than sampling from the composite distribution $p'(z|x, y)$ directly. To obtain unbiased estimates of the KL divergence under this sampling scheme, we employ importance weighting that accounts for the distribution mismatch.

**When sampling from the prior distribution:** The token-level importance sampling ratio is:

$$w_t = \frac{p'(z_t|z_{<t}, x, y)}{p_\phi(z_t|z_{<t}, x)} = \frac{1}{2} + \frac{1}{2}r_t \tag{22}$$

Since we are estimating $D_{\mathrm{KL}}(p'\|p_\phi)$ (i.e., forward KL), we apply the Bregman divergence formula for this direction: $w \log w - (w - 1)$. This gives:

$$\begin{aligned} D_{\mathrm{KL}}^{\mathrm{prior}} &= w_t \log w_t - (w_t - 1) \\ &= \left(\frac{1}{2} + \frac{1}{2}r_t\right)\log\left(\frac{1}{2} + \frac{1}{2}r_t\right) - \left(\frac{1}{2}r_t - \frac{1}{2}\right) \end{aligned} \tag{23}$$

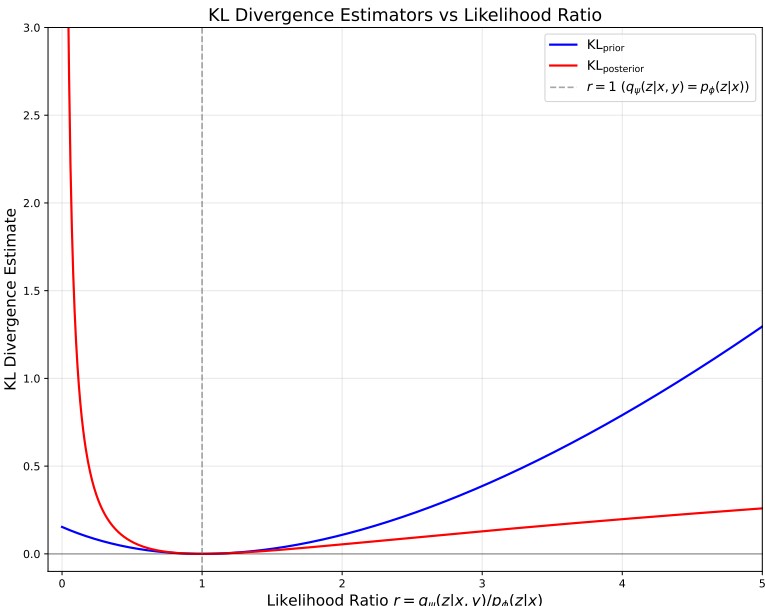

*Figure 6.* Comparison of KL divergence estimators under different sampling strategies.

**When sampling from the posterior distribution:** The token-level importance sampling ratio is:

$$w_t = \frac{p'(z_t|z_{<t}, x, y)}{q_\psi(z_t|z_{<t}, x, y)} = \frac{1}{2r_t} + \frac{1}{2} \tag{24}$$

To estimate $D_{\text{KL}}(p'\|p_\phi)$, we need $\log p' - \log p_\phi$. Using importance sampling from $q_\psi$:

$$
\begin{aligned}
D_{\text{KL}}^{\text{posterior}} &= w_t \log \frac{p'(z_t|z_{<t}, x, y)}{p_\phi(z_t|z_{<t}, x)} + (w_t - 1) \\
&= \left( \frac{1}{2r_t} + \frac{1}{2} \right) \log \left( \frac{1}{2} + \frac{1}{2}r_t \right) + \left( \frac{1}{2r_t} - \frac{1}{2} \right)
\end{aligned}
\tag{25}
$$

Note the sign difference in the correction term: $-(w_t - 1)$ for prior sampling versus $+(w_t - 1)$ for posterior sampling. Both forms ensure non-negativity through the Bregman divergence property.

Figure 6 shows the behavior of both estimators across different likelihood ratios. When $r > 1$, the posterior assigns higher probability to the sample than the prior, indicating reasoning traces well-aligned with target answers. When $0 < r < 1$, the prior assigns higher probability, representing more exploratory reasoning paths. Our hybrid sampling strategy leverages this complementary behavior to ensure stable optimization across diverse reasoning patterns.

### A.3. Relationship with On-Policy Distillation

Recent work on On-Policy Distillation (OPD) (Lu & Lab, 2025; Hübotter et al., 2026; Penaloza et al., 2026) proposes a hybrid training paradigm that bridges reinforcement learning and supervised fine-tuning. While reinforcement learning optimizes self-sampled trajectories using sparse rewards, supervised fine-tuning provides dense token-level supervision but is typically performed off-policy. OPD integrates these two paradigms by imposing token-level teacher supervision directly on trajectories sampled from the student policy.

Let $p_\phi(z|x)$ denote the student policy and $T(z|x)$ the teacher distribution. OPD optimizes the following objective:

$$\mathcal{L}_{\text{OPD}} = \mathbb{E}_{z \sim p_\phi(z|x)} \left[ D_{\text{KL}} \left( p_\phi(\cdot|x, z_{<t}) \| T(\cdot|x, z_{<t}) \right) \right]. \tag{26}$$

Although OPD provides dense token-level supervision under on-policy sampling, several studies (Li et al., 2026; Fu et al., 2026) have highlighted critical limitations. These include the support mismatch between student-generated trajectories and

the fixed teacher distribution, as well as error accumulation in long-horizon reasoning caused by progressively miscalibrated teacher probabilities on shifted prefixes.

Unlike OPD, which relies on a static external teacher, CoVRL introduces a coupled variational formulation incorporating a prior $p_\phi(z|x)$ and a posterior $q_\psi(z|x, y)$. The prior governs inference-time generation, whereas the posterior provides answer-conditioned training signals. To bridge them, we define a symmetric composite distribution:

$$p'(z_t|z_{<t}, x, y) = \frac{1}{2}p_\phi(z_t|z_{<t}, x) + \frac{1}{2}q_\psi(z_t|z_{<t}, x, y). \tag{27}$$

This construction establishes a bidirectional coupling between the two distributions. The posterior provides answer-guided supervisory signals to refine the prior, while the evolving prior dynamically shapes the training distribution for the posterior. Consequently, the posterior continuously adapts to the current state of the prior rather than acting as a fixed oracle. This data-dependent coupling preserves the advantages of dense token-level supervision during on-policy sampling while circumventing the reliance on a static external teacher.

This coupling mechanism, alongside our hybrid sampling strategy, can be interpreted through the lens of the Jensen–Shannon divergence (JSD), formally defined as:

$$\mathrm{JSD}(p_\phi\|q_\psi) = \frac{1}{2}D_{\mathrm{KL}}(p_\phi\|p') + \frac{1}{2}D_{\mathrm{KL}}(q_\psi\|p') \tag{28}$$

By utilizing the composite distribution $p' = \frac{1}{2}(p_\phi + q_\psi)$ as a shared variational reference, our formulation yields a stabilizing effect remarkably similar to JSD. Theoretically, this symmetric coupling fundamentally mitigates the unidirectional optimization pathologies in standard OPD.

Despite this structural resemblance, the KL divergence term in Eq. 8 cannot be directly replaced by the standard JSD. In our context, the prior and posterior serve fundamentally different purposes. Our ultimate objective is still asymmetric: we aim to maximize the capability of answering correctly during inference without answer guidance, rather than forcing the prior and posterior into a perfect, symmetric compromise. Consequently, our objective is to facilitate a directed information flow from the posterior to the prior, rather than penalizing both equally toward the composite distribution $p'$.

Nevertheless, by applying specific modifications to Eq. 8, we can derive a strictly symmetric formulation grounded in variational reward maximization and the Jensen–Shannon divergence. Specifically, instead of directly optimizing the asymmetric KL divergence, we can define the overall target distribution as our composite distribution $p'(z|x, y) = \frac{1}{2}p_\phi(z|x) + \frac{1}{2}q_\psi(z|x, y)$ and seek to maximize the log-expected reward under this composite distribution, $\log \mathbb{E}_{p'}[e^{R(z)}]$. To operationalize this, we introduce a hybrid proposal distribution $p_{\mathrm{hybrid}}(z|x, y)$ that samples from the prior $p_\phi(z|x)$ with probability $\alpha$, and from the posterior $q_\psi(z|x, y)$ with probability $1 - \alpha$. Mathematically, this is expressed as:

$$p_{\mathrm{hybrid}}(z|x, y) = \alpha p_\phi(z|x) + (1 - \alpha)q_\psi(z|x, y). \tag{29}$$

By employing this hybrid sampling strategy with $\alpha = 0.5$, we can derive the corresponding ELBO:

$$
\begin{aligned}
\log \mathbb{E}_{p'}\left[e^{R(z)}\right] &= \frac{1}{2}\log \mathbb{E}_{p'}\left[e^{R(z)}\right] + \frac{1}{2}\log \mathbb{E}_{p'}\left[e^{R(z)}\right] \\
&= \frac{1}{2}\log \int p_\phi(z|x) \frac{p'(z|x, y)}{p_\phi(z|x)} e^{R(z)}\, dz + \frac{1}{2}\log \int q_\psi(z|x, y) \frac{p'(z|x, y)}{q_\psi(z|x, y)} e^{R(z)}\, dz \\
&= \frac{1}{2}\log \mathbb{E}_{p_\phi}\left[\frac{p'(z|x, y)}{p_\phi(z|x)} e^{R(z)}\right] + \frac{1}{2}\log \mathbb{E}_{q_\psi}\left[\frac{p'(z|x, y)}{q_\psi(z|x, y)} e^{R(z)}\right] \\
&\geq \frac{1}{2}\mathbb{E}_{p_\phi}\left[R(z) + \log \frac{p'(z|x, y)}{p_\phi(z|x)}\right] + \frac{1}{2}\mathbb{E}_{q_\psi}\left[R(z) + \log \frac{p'(z|x, y)}{q_\psi(z|x, y)}\right] \quad \text{(Jensen)} \\
&= \left(\frac{1}{2}\mathbb{E}_{p_\phi}[R(z)] + \frac{1}{2}\mathbb{E}_{q_\psi}[R(z)]\right) - \left(\frac{1}{2}D_{\mathrm{KL}}(p_\phi\|p') + \frac{1}{2}D_{\mathrm{KL}}(q_\psi\|p')\right) \\
&= \mathbb{E}_{p_{\mathrm{hybrid}}}[R(z)] - \mathrm{JSD}(p_\phi\|q_\psi).
\end{aligned}
\tag{30}
$$

This derivation reveals that maximizing the variational reward under a symmetrically sampled composite distribution naturally induces a JSD-regularized objective. While our current CoVRL objective maintains an asymmetric design to

explicitly prioritize the inference-time generation capability of the prior, exploring this strictly symmetric JSD formulation presents a highly promising avenue for future work. It would be particularly valuable in task scenarios where the performance, calibration, and continuous co-evolution of both the prior and the posterior distributions are of equal importance.

## B. Comparison Among Existing Approaches

We compare the gradient estimators of various verifier-free reasoning methods:

$$
\begin{aligned}
\nabla_\theta J_{\text{JLB}} &= \mathbb{E}_{z \sim \pi_\theta(z|x)} \big[ \log \pi_\theta(y^*|x, z) \nabla_\theta \log \pi_\theta(z|x) + 1 \cdot \nabla_\theta \log \pi_\theta(y^*|x, z) \big] \\
\nabla_\theta J_{\text{LaTRO}} &= \mathbb{E}_{z \sim \pi_\theta(z|x)} \big[ (\log \pi_\theta(y^*|x, z) - \beta \log \frac{\pi_\theta(z|x)}{\pi_{\text{ref}}(z|x)}) \nabla_\theta \log \pi_\theta(z|x) + 1 \cdot \nabla_\theta \log \pi_\theta(y^*|x, z) \big] \\
\nabla_\theta J_{\text{VeriFree}} &= \mathbb{E}_{z \sim \pi_\theta(z|x)} \big[ \pi_\theta(y^*|z, x) \nabla_\theta \log \pi_\theta(z|x) + \pi_\theta(y^*|z, x) \nabla_\theta \log \pi_\theta(y^*|x, z) \big] \\
\nabla_\theta J_{\text{RLPR}} &= \mathbb{E}_{z \sim \pi_\theta(z|x)} \big[ \frac{\sum(\{p_i | t_i \in y^*\})}{|y^*|} \nabla_\theta \log \pi_\theta(y^*, z|x) \big] \\
\nabla_\theta J_{\text{RAVR}} &= \mathbb{E}_{z \sim \pi_\theta(z|x, y^*)} \big[ R(z) \nabla_\theta \log \pi_\theta(z|x) \big] + \nabla_\theta R(z) \times D_{\text{KL}}(p(z|x, y^*) \| p(z|x)), \text{ where} \\
& \qquad R(z) = \max(0, \log \pi_\theta(y^*|z, x) - \mathbb{E}_{z' \sim \pi_\theta(z|x)}[\log \pi_\theta(y^*|x, z')])
\end{aligned}
\tag{31}
$$

All methods aim to improve reasoning without external verifiers but differ in their approach. JLB (Tang et al., 2025) uses log-probability as a reward with fixed answer term weighting. LaTRO (Chen et al., 2024) incorporates KL regularization between policy and reference models. VeriFree (Zhou et al., 2025a) uses the probability instead of log-probability as reward and also uses it as the weight term for NLL loss. RLPR (Yu et al., 2025b) optimizes joint answer-reasoning probability using token-level probabilities as rewards. RAVR (Lin et al., 2025) samples from posterior distributions and optimizes the corresponding variational lower bound. The core differences among these methods lie in the sampling distribution, reward design, and weighting coefficients of the NLL loss terms.

In contrast, our CoVRL introduces a fundamentally different approach through hybrid sampling and composite distribution optimization. The gradient combines three key components: (1) importance-weighted policy gradient on the composite distribution $p'(z|x, y)$ for efficient exploration-exploitation balance, (2) selective NLL loss applied only to high-quality samples ($\mathbb{I}[\hat{A} > 0]$), and (3) explicit KL regularization to ensure transferability from training to inference.

$$
\begin{aligned}
\nabla J_{\text{CoVRL}} &= \mathbb{E}_{z \sim p_{\text{hybrid}}(z|x, y)} \Big[ \frac{p'(z|x, y^*)}{p_{\text{hybrid}}(z|x, y^*)} \times \log p_\theta(y^*|x, z) \nabla \log p'(z|x, y) \\
& \qquad + \mathbb{I}[\hat{A} > 0] \nabla \log p_\theta(y^*|x, z) \Big] - \nabla D_{\text{KL}}(p'(z|x, y^*) \| p_\phi(z|x))
\end{aligned}
\tag{32}
$$

CoVRL leverages both question-only and answer-guided sampling through $p_{\text{hybrid}}(z|x, y)$, enabling more effective training while maintaining inference compatibility.

## C. Evaluation Implementation

For non-multiple-choice questions, we use the math-verify library to directly parse the standard answers. For multiple-choice questions, we standardize all datasets into a unified multiple-choice format. When parsing model responses, we employ different strategies based on the question type:

**Non-multiple-choice questions:** We first use the default parsing method of the math-verify library to extract results. If no parsing result is obtained, we wrap the answer with `$$` to parse it as a LaTeX expression. This approach handles cases where the model outputs `<answer>latex expression</answer>` without proper LaTeX delimiters. If there is still no valid parsing result, we perform exact string matching as a fallback.

**Multiple-choice questions:** When parsing standard answers, we include both the option labels (e.g., A, B, C, D) and the option contents as potential matches, since the model sometimes outputs the option content instead of the option label. When parsing model outputs, we first attempt to match the option format. If option matching fails, we try to parse the response as LaTeX format. If LaTeX parsing also fails, we perform exact string matching detection.

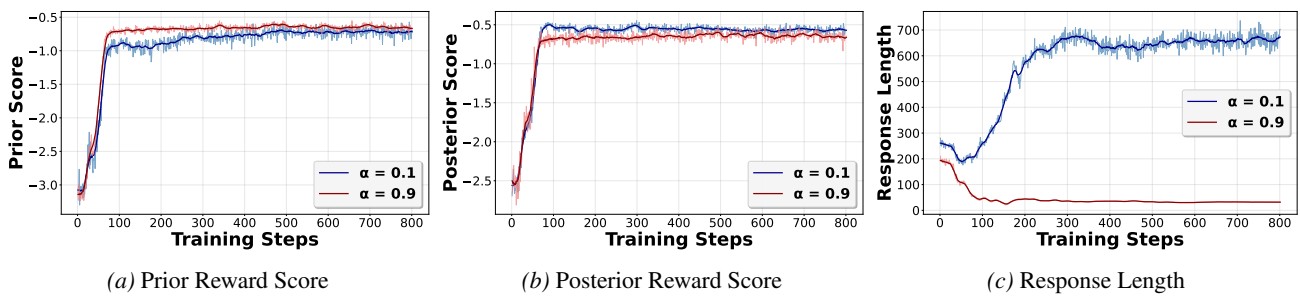

*(a)* Prior Reward Score  *(b)* Posterior Reward Score  *(c)* Response Length

*Figure 7.* Training dynamics comparison between different prior sampling probabilities ($\alpha = 0.1$ vs $\alpha = 0.9$).

*Table 4.* Ablation study on regularization coefficients and reward functions.

| Configuration | General | | | Mathematical | | | | | | |
| | GPQA Avg@4 | MMLU-Pro Avg@1 | TheoremQA Avg@2 | AIME'24 Avg@16 | AQuA Avg@4 | CARP-EN Avg@2 | MATH-500 Avg@4 | Minerva Avg@4 | SAT-Math Avg@32 | Overall |
|---|---|---|---|---|---|---|---|---|---|---|
| *Regularization Coefficients* | | | | | | | | | | |
| Default | 30.4 | 46.5 | 36.3 | 7.5 | 77.3 | 65.1 | 66.3 | 25.5 | 97.1 | 50.2 |
| $\lambda_{KL} = 0.1$ | 31.3 | 32.5 | 20.7 | 0.3 | 35.1 | 33.8 | 18.1 | 11.9 | 62.6 | 27.4 |
| $\lambda_{NLL} = 0.1$ | 31.2 | 42.5 | 33.1 | 4.2 | 69.0 | 49.6 | 59.5 | 23.5 | 89.6 | 44.7 |
| *Reward Functions* | | | | | | | | | | |
| Log-Probability | 30.4 | 46.5 | 36.3 | 7.5 | 77.3 | 65.1 | 66.3 | 25.5 | 97.1 | 50.2 |
| Probability | 31.4 | 46.9 | 35.8 | 6.3 | **78.2** | **65.3** | **67.6** | 27.0 | 94.9 | 50.4 |
| Probability Sum | 28.1 | 45.9 | **37.2** | 7.3 | 74.8 | 65.0 | 68.7 | 24.9 | 95.1 | 49.7 |
| Log-Probability Sum | **31.5** | **47.7** | 36.7 | **7.1** | 76.8 | 65.1 | 67.1 | **27.4** | **96.0** | **50.6** |

## D. Training Dynamics of Different Sampling Ratios

Figure 7 illustrates the distinct training dynamics under different sampling ratios. As shown in the figure, $\alpha = 0.9$ achieves higher prior reward scores, whereas $\alpha = 0.1$ obtains higher posterior reward scores. This is consistent with our expectations: models trained with more prior sampling naturally perform better under prior evaluation, while those trained with more posterior sampling excel under posterior evaluation. In addition, response lengths increase during training when $\alpha = 0.1$ but decrease when $\alpha = 0.9$. This reflects the different roles of the prior and posterior distributions in **exploration** and **exploitation**.

In particular, when $\alpha = 0.9$, the KL term in Eq. 8 mainly optimizes the teacher distribution conditioned on student prefixes. This primarily encourages exploitation, i.e., learning to match the teacher distribution over answers based on the student's existing generations. As a result, the student distribution is encouraged to cover the answer more effectively given the reference answer and the current solution state, thereby minimizing the KL divergence to the teacher distribution, which naturally leads to shorter answers.

In contrast, when $\alpha = 0.1$, the KL term mainly optimizes the student distribution conditioned on teacher prefixes. This encourages exploration by allowing the model to build on the stronger generations provided by the teacher. Under this setting, the KL term encourages the student distribution to explore possible states induced by teacher prefixes rather than directly converging to the answer, which results in longer answers. As shown in Figure 5, both mechanisms are important, and balancing exploration and exploitation leads to the best overall performance.

## E. Influence of Loss Components

We conduct comprehensive ablation studies to understand the impact of key components in our CoVRL framework, including regularization coefficients and reward function formulations. The results are presented in Table 4.

**KL regularization** The results show that reducing the KL divergence coefficient to 0.1 significantly degrades performance across all benchmarks, with overall accuracy dropping to 27.4%. This performance degradation stems from training instability when the KL regularization is insufficient. We observe substantial increases in KL divergence during training, indicating significant deviation between the prior and posterior distributions. This leads to both training-inference mismatch

*Table 5.* Performance comparison of reasoning models across different benchmarks.

| Model | General | | | Mathematical | | | | | | Overall |
|---|---|---|---|---|---|---|---|---|---|---|
| | GPQA @4 | MMLU-Pro @1 | TheoremQA @2 | AIME'24 @16 | AQuA @4 | CARP-EN @2 | MATH-500 @4 | Minerva @4 | SAT-Math @32 | |
| Qwen3-8B | 30.0 | 58.5 | 43.8 | 17.9 | 81.3 | 67.8 | 81.5 | 41.0 | 81.7 | 55.9 |
| + CoVRL | $41.8_{+11.8}$ | $63.2_{+4.7}$ | $47.4_{+3.6}$ | $24.3_{+6.4}$ | $81.2_{-0.1}$ | $65.6_{-2.2}$ | $82.2_{+0.7}$ | $37.3_{-3.7}$ | $96.1_{+14.4}$ | $59.9_{+4.0}$ |
| Qwen3-14B | 39.6 | 68.0 | 50.7 | 20.2 | 82.9 | 68.4 | 83.6 | 44.0 | 97.2 | 61.6 |
| + CoVRL | $46.1_{+6.5}$ | $67.6_{-0.4}$ | $50.8_{+0.1}$ | $24.8_{+4.6}$ | $84.6_{+1.7}$ | $65.3_{-3.1}$ | $83.2_{-0.4}$ | $40.6_{-3.4}$ | $99.1_{+1.9}$ | $62.5_{+0.9}$ |

*Table 6.* Time breakdown (in seconds) per training step.

| Method | Rollout | Reward | Advantage | Actor Update | Old Logprobs | Overall |
|---|---|---|---|---|---|---|
| GRPO | 21.70 | 0.36 | 0.03 | 5.93 | 2.30 | 30.32 |
| CoVRL | 21.94 | 0.51 | 0.03 | 10.60 | 3.01 | 36.09 |

issues and training instability, as we are essentially performing off-policy optimization with an increasing distribution shift between training and inference.

**NLL loss** In contrast, the model appears less sensitive to changes in the NLL loss coefficient. When reducing the NLL coefficient to 0.1, performance decreases moderately to 44.7%. We attribute this resilience to the fact that the RL term and NLL loss optimize essentially the same objective, both of which aim to improve answer prediction quality. The NLL loss primarily trains the model's ability to summarize reasoning and produce final answers.

**Reward function formulation** We examine different reward formulations for our variational framework, specifically focusing on **(1) Length normalization**: comparing averaging over sequence length versus unnormalized probability sums; and **(2) Logarithmic transformation**: examining whether to use log-probabilities or raw probabilities as reward signals. The results demonstrate that all reward formulations achieve remarkably similar overall performance, with variations of less than 1 percentage point (49.7% to 50.6%). This consistency indicates that our CoVRL framework is robust to various reward formulations.

## F. Improvement on Reasoning Model

In addition to evaluating CoVRL on base models, we investigate its effectiveness when applied to models that have already undergone reasoning-specific training. Specifically, we apply CoVRL to Qwen3-8B and Qwen3-14B reasoning models (Yang et al., 2025). Since reasoning models naturally generate longer chain-of-thought processes, we increase the maximum response length from 2048 to 4096 tokens during training to avoid excessive clipping of reasoning traces. This adjustment ensures that the models can fully express their reasoning processes without being frequently truncated.

As shown in Table 5, CoVRL continues to improve overall performance when applied to reasoning models. For Qwen3-8B, the overall improvement is 4.0% (from 55.9% to 59.9%), while for Qwen3-14B, the improvement is 0.9% (from 61.6% to 62.5%). We observe mixed results across individual benchmarks, with notable improvements on some tasks (e.g., GPQA +11.8% and SAT-Math +14.4% for Qwen3-8B; GPQA +6.5% and AIME'24 +4.6% for Qwen3-14B) but slight degradation on others (e.g., Minerva -3.7% for Qwen3-8B, CARP-EN -3.1% for Qwen3-14B). We attribute this variation to two factors: (1) these reasoning models already possess strong baseline capabilities, leaving limited room for improvement; and (2) our training data was not specifically filtered or curated for these high-performance models, potentially introducing noise that affects certain benchmarks. Nevertheless, the consistent improvements in overall performance demonstrate the robustness and effectiveness of CoVRL, even when applied to models with already sophisticated reasoning capabilities.

## G. Computation Cost Analysis

Compared with GRPO, CoVRL incurs comparable rollout generation cost under the same batch size. The computational overhead primarily arises from two stages: (1) the actor update, where CoVRL optimizes both distributions, and (2) the

*Table 7.* Reasoning process evaluation on 100 randomly sampled MMLU-Pro questions. Base denotes Qwen2.5-7B-Base. Scores are on a 1-5 scale and reported as mean(std) over 8 judge queries. Fact., Val., Coh., and Util. denote factuality, validity, coherence, and utility, respectively.

| Evaluator | Model | Fact. | Val. | Coh. | Util. |
|---|---|---|---|---|---|
| GPT-OSS | Base | 2.56(.04) | 2.82(.04) | 3.72(.03) | 2.68(.04) |
|  | + CoVRL | **2.79(.01)**$_{+0.23}$ | **3.01(.04)**$_{+0.19}$ | **3.77(.04)**$_{+0.05}$ | **2.93(.04)**$_{+0.25}$ |
| GPT-5.2 | Base | 2.55(.03) | 2.73(.02) | 3.83(.02) | 2.90(.03) |
|  | + CoVRL | **2.72(.04)**$_{+0.17}$ | **2.83(.02)**$_{+0.10}$ | **3.87(.02)**$_{+0.04}$ | **3.05(.02)**$_{+0.15}$ |
| Claude-Sonnet-4.6 | Base | 2.76(.04) | 2.69(.03) | 3.16(.03) | 2.48(.03) |
|  | + CoVRL | **3.01(.03)**$_{+0.25}$ | **2.87(.02)**$_{+0.18}$ | **3.26(.05)**$_{+0.10}$ | **2.85(.03)**$_{+0.37}$ |

computation of old log-probabilities. Regarding GPU memory, peak usage remains largely unchanged by using vLLM for memory-efficient generation and maintaining a fixed micro-batch size via gradient accumulation. To quantify this overhead, we measure the time breakdown for a single training step using the Qwen2.5-7B-Base model on $4 \times 8$ GPUs with a global batch size of 192 and a maximum response length of 2048, as summarized in Table 6.

The empirical results align with our analysis: although the actor update cost approximately doubles, rollout generation dominates the total step time. Consequently, the end-to-end overhead is relatively small, increasing from 30.32 to 36.09 seconds. Furthermore, under a fully on-policy implementation, the computation of old log-probabilities can be merged with the actor-update forward pass, eliminating this specific overhead relative to standard GRPO.

## H. Reasoning Process Evaluation

We further evaluate the quality of intermediate reasoning traces. Following Lee & Hockenmaier (2025), we consider four dimensions: **factuality**, **validity**, **coherence**, and **utility**. Factuality measures whether reasoning steps are grounded in reliable sources. Validity measures whether reasoning steps are free from logical errors. Coherence measures whether each step is supported by previous steps. Utility measures whether each step helps reach the final answer. We randomly sample 100 questions from MMLU-Pro and use multiple open- and closed-source LLMs as judges. Each judge scores the reasoning traces of Qwen2.5-7B-Base before and after CoVRL training on a 1-5 scale. Each judge is queried 8 times per sample, and we report the mean and standard deviation.

Table 7 shows that CoVRL consistently improves all four dimensions across all evaluators. This suggests that our method improves not only final-answer accuracy but also the quality of intermediate reasoning traces. While LLM-as-judge evaluation cannot replace human expert annotation, the consistent gains across multiple independent judges provide additional evidence that CoVRL produces more reliable reasoning processes.

