# OpenReview forum: "Coupled Variational Reinforcement Learning for Language Model General Reasoning"
_ICML.cc/2026/Conference — ICML 2026 regular_

### Official Review · Reviewer_NmWc · 2026-02-14

**Soundness:** 2
**Presentation:** 2
**Significance:** 2
**Originality:** 3
**Overall Recommendation:** 4
**Confidence:** 3

**Summary:**

This paper proposes Coupled Variational Reinforcement Learning (CoVRL), a verifier-free RL framework for improving language model reasoning without relying on externally verifiable rewards. It studies how to couple question-conditioned prior sampling with answer-guided posterior sampling to improve exploration efficiency and reasoning–answer coherence; this paper addresses this through a variational objective with a composite distribution and hybrid sampling strategy.

**Compliance With Llm Reviewing Policy:**

Affirmed.

**Final Justification:**

See above. But note that I have a relatively low confidence score.

**Key Questions For Authors:**

See weaknesses

**Limitations:**

It would also be valuable to discuss the limitations of the core assumption that the probability of generating the reference answer is a meaningful reward signal. For example, a model may assign high probability to the reference answer without producing coherent or correct reasoning, or it may generate a correct answer in a different but equivalent format and receive a low reward. Moreover, this reward design may encourage answer memorization or shallow pattern matching rather than genuine reasoning improvement. A clearer discussion of these potential limitations would strengthen the paper.

**Strengths And Weaknesses:**

**Warning: I have a relatively weak theoretical background, so my evaluation is mainly based on the empirical part of this work. This is also why my confidence score is 3.**

+ Strengths
  + I personally appreciate the high-level research problem. In many realistic settings, we only have reference answers but no reliable verifier that can provide verifiable rewards (e.g., it may be hard to implement a Python-based verifier that is not prone to false negatives). Understanding how to perform RL under such constraints is an important and practical question.
  + The idea appears novel to me. However, given my limited theoretical background, I am not fully confident in assessing the novelty and rigor of the theoretical contributions.

+ Weaknesses
  + Although the paper assumes that no verifier is available, in the experimental setup it is still possible to construct approximate verifiers (e.g., Python-based tools such as math-verify) and run standard GRPO. Even if such verifiers are imperfect, what ultimately matters is final performance. Therefore, I believe it would be valuable to include such a baseline for comparison. If the authors disagree, a more explicit discussion would be helpful.

  + The paper lacks some important training details. For example, how many total training steps were used, and at which step was each method evaluated? It would be helpful to even provide evaluation curves over training steps to better understand convergence behavior (for example, is every baseline converged?) and training stability.

  + The experiments in Table 2 provide limited insight into the strength of training signals. Given recent findings on spurious rewards (e.g., https://arxiv.org/abs/2506.10947), improvements on commonly used benchmarks such as AIME and other math datasets can sometimes be achieved relatively easily. Therefore, it would be important to compare against stronger or more carefully designed baselines to ensure that the gains are meaningful.

---

> ### Author Rebuttal · Authors · 2026-03-31
>
> Thank you for acknowledging the novelty and soundness of our work.  We would like to address the reviewer’s concerns as follows.
>
> **Weakness1:**
>
> Thank you for this important suggestion. We agree that, for some tasks in mathematics and coding, robust rule-based verifiers are available. However, for the broader general reasoning tasks targeted in this work, such verifiers are often difficult to construct. Even when the final answer is relatively short, tasks in areas such as physics or chemistry may not admit a simple rule-based verifier with sufficiently high accuracy. For example, a chemistry problem may require generating a reaction equation such as “2 CH₃COOH + CaC₃H₇O₆P → Ca(CH₃COO)₂ + C₃H₉O₆P”, for which there may be no readily available verification system capable of reliably determining correctness. In our work, we avoid the difficulty of constructing robust verifiers by treating the model’s probability of generating the target answer as the reward signal under a variational inference formulation.
>
> **Weakness2:**
>
> We provide the training curves in Figure 4 and discuss the training dynamics in Section 3.3. For both our method and all baselines, we train the model for exactly one epoch (801 steps, as shown in Figure 4), and we consistently use the final checkpoint for evaluation. Given the relatively long training schedule and the training dynamics observed in Figure 4, the models should have largely converged. We will add these training details and other necessary information to the paper.
>
> **Weakness3:**
>
> In Table 1, we compare against several strong baselines and observe a 2.3% overall improvement. In addition, Appendix F further validates the effectiveness of our method on models that have already undergone reasoning-specific training, such as Qwen3-8B and Qwen3-14B. The results show that our method is still able to improve overall performance on these strong starting points, even without specially curated data. Taken together, these results provide evidence for the effectiveness of our training signal.
>
> **Limitations:**
>
> The issue you raised is indeed real, and it is precisely one of the challenges that CoVRL is designed to address, which we refer to as *trace–answer incoherence*. For the first case you mentioned—where the model generates a correct answer in a different but equivalent form and therefore receives a low reward—we mitigate this issue through the introduction of the posterior distribution and hybrid sampling. When sampling from the prior distribution (i.e., conditioned only on the question), the model does not have access to the reference answer, and may therefore produce an answer that is semantically correct but differs in surface form, leading to a low reward. In such cases, however, the underlying reasoning trace may still be largely correct. As a result, the KL term in Eq. 8, measured with respect to the posterior distribution, can remain relatively small and still provide a useful training signal. Moreover, because our method also samples from the posterior distribution, the model is exposed to the reference answer during part of training, which substantially reduces the probability of this form-mismatch issue.
>
> For the second case you mentioned—where the model assigns high probability to the reference answer despite producing incoherent or incorrect reasoning—our method mainly addresses it through the KL regularization term in Eq. 8. When sampling from the posterior distribution, the model could in principle fabricate an inconsistent reasoning trace that still leads to the correct answer. However, such a trace is typically unlikely to be generated without answer guidance, and therefore incurs a relatively large KL penalty. When sampling from the prior distribution, we make the same standard assumption as in RLVR-style reasoning training: reasoning traces that lead to the correct answer are generally useful training signals. Even if the model arrives at the answer partly by guessing, such traces may still be penalized through inconsistency with the posterior distribution, again via the KL term.
>
> Thank you for pointing this out. We will add a more detailed discussion of these cases in the revised version.

---

> > ### Author Rebuttal · Reviewer_NmWc · 2026-03-31
> >
> > Thanks for the rebuttal. I would like to keep my score of 4 because I don't have a very high confidence score.

---

### Official Review · Reviewer_Wn7g · 2026-03-04

**Soundness:** 3
**Presentation:** 2
**Significance:** 3
**Originality:** 3
**Overall Recommendation:** 3
**Confidence:** 4

**Summary:**

This paper proposed Coupled Variational Reinforcement Learning (CoVRL), which frames reasoning training as a variational optimization problem that couples prior and posterior distribution. CoVRL models the reasoning process as a latent variable and formulates the reasoning training problem as one of variational inference. In existing validator-free reinforcement learning, reasoning is sampled only based on the question, which leads to low exploration efficiency and inconsistency between reasoning and the final answer. To address this issue, CoVRL achieves the joint modeling of reasoning generation and answer information by coupling a prior distribution (conditioned only on question) and a posterior distribution (conditioned on both question and answer).

The authors construct a composite distribution that fuses the prior and posterior, and propose a hybrid sampling strategy to randomly switch between the two generation modes during training. The model is jointly optimized by maximizing the ELBO, which includes a reconstruction term that promotes correct answer generation and a KL regularization term that ensures consistency between training and inference. Furthermore, the method combines importance sampling and policy optimization (e.g., GRPO) for efficient training. Empirically, CoVRL is evaluated on a variety of general reasoning and mathematical reasoning benchmarks. The results demonstrate that CoVRL achieves significant performance improvements over base models and outperforms existing validator-free reinforcement learning methods.

**Compliance With Llm Reviewing Policy:**

Affirmed.

**Final Justification:**

The authors use multiple LLMs to score the reasoning processes. Although CoVRL shows some improvement over baselines, the scores remain modest: above 3 only on Coherence, and around 2.5–2.8 on other key dimensions (Factuality, Validity, and Utility) on a **1–5** scale. Given that the quality of the reasoning process is central to the paper’s claim, these relatively low scores raise doubts about whether the method genuinely improves the correctness or logical soundness of the reasoning.

I will keep my score.

**Key Questions For Authors:**

1. Discuss the limitations of CoVRL.

2. In Section 3.4, the choice of $\alpha$ values {$0.1, 0.5, 0.9$} for sampling from the prior distribution is not sufficiently justified. Could the authors provide a more fine-grained analysis of how $\alpha$ affects the model performance?

3. In Table 3, what causes the performance degradation of 'Math Only' on CARP-EN?

4. For some small test sets, multiple runs are conducted to assess stability, $N=$ {$ 2,4,16,32,...$ } . I suggest reporting the standard deviation or variance of the results.

5. The naming should be made consistent, e.g., “Qwen2.5-7B-Base” & “Qwen2.5-7B-base” in Section 3.1.

**Limitations:**

I suggest that the authors discuss the limitations of CoVRL more explicitly. In particular, it would be helpful to clarify the scenarios in which training relies on ground-truth answers, as well as how potential logical flaws in the reasoning process can be evaluated.

**Strengths And Weaknesses:**

Strengths:

1. Improving reasoning quality and consistency in a verifier-free setting has practical potential for reducing training costs and broadening the range of applications.

2. The method introduces some novelty by integrating variational inference with reinforcement learning and incorporating prior–posterior coupling.

3. CoVRL is evaluated on several reasoning and mathematical benchmarks, and the results support the main claims of performance improvement.

4. The paper is well-written, and the problem tackled seems of great interest.

Weaknesses:

1. The paper does not discuss the limitations of the proposed method.

2. The paper lacks sufficient experimental details and analysis of parameter sensitivity.

3. Lacks evaluation of the correctness of the reasoning process.

---

> ### Author Rebuttal · Authors · 2026-03-31
>
> Thank you for acknowledging the novelty of our work.  We would like to address the reviewer’s concerns as follows.
>
> 1. **Limitations:**
>
> A primary limitation of CoVRL is the additional computation introduced by the posterior distribution. In particular, CoVRL requires two forward passes in both the actor-update stage and the old log-probability computation stage, as both the prior and posterior distributions need to be evaluated in each stage. As a result, these stages are more expensive than in standard RL training. However, this does not lead to a 2× increase in end-to-end training time in practice, since rollout typically dominates the overall cost. Under a unified setup with Qwen2.5-7B-Base + FSDP2 + vLLM, on the same 4 × 8 GPUs, with batch size = 192 and maximum response length = 2048, the measured time breakdown is as follows:
>
> | Setting/Stage | Generated batch | Compute reward | Compute advantage | Update actor | Compute old logprobs | Overall |
> | --- | --- | --- | --- | --- | --- | --- |
> | CoVRL | 21.94 sec. | 0.51 sec. | 0.03 sec. | 10.60 sec. | 3.01 sec. | 36.09 sec. |
> | GRPO | 21.70 sec. | 0.36 sec. | 0.03 sec. | 5.93 sec. | 2.30 sec. | 30.32 sec. |
>
> The actor-update cost is approximately doubled, and the per-step training time increases by about 19.0% compared with GRPO. We agree that this computational trade-off should be stated more explicitly, and we will add a discussion of  limitations to the revised paper.
>
> 1. **Experiment details, hyper-parameters influence and  analysis of $\alpha$ :**
>
> We currently describe our training and evaluation settings in Section 3.1 and Appendix C. Regarding hyper-parameter sensitivity, we investigate different components of our method in Appendix D and Appendix E, including different loss terms, reward function implementations, and the hybrid sampling ratio α. Overall, we find that the natural default choices derived in the main paper generally work best, which suggests that the components introduced in the CoVRL framework are all important. As for the reward function implementation, we find that performance is not sensitive to the specific design choice: different length normalization strategies and logarithmic transformations achieve competitive performance. We will add more hyper-parameter details and analysis in the revised version.
>
> Regarding the choice of $\alpha \in \{0.1,0.5,0.9\}$, we selected these values to examine mainly prior-sampled, balanced, and mainly posterior-sampled regimes, respectively, so as to better understand the roles of question-only and answer-guided generation. In particular, when $α = 0.1$, the KL term in Eq. 8 mainly optimizes the teacher distribution conditioned on student prefixes, which primarily encourages exploitation, i.e., learning to match the teacher distribution over answers based on the student’s existing generations. In contrast, when $α = 0.9$, the KL term mainly optimizes the student distribution conditioned on teacher prefixes, which encourages exploration by allowing the model to build on the stronger generations provided by the teacher. As shown in Figure 7, these two settings exhibit different optimization behaviors. We agree that a more fine-grained analysis would be helpful, and we therefore additionally evaluate more α values:
>
> | Result/Alpha | Alpha=0.1 | Alpha=0.3 | Alpha=0.5 | Alpha=0.7 | Alpha=0.9 |
> | --- | --- | --- | --- | --- | --- |
> | Overall Performance | 0.455 | 0.492 | 0.502 | 0.303 | 0.295 |
>
> We observe that the balanced strategy ($α = 0.5$) performs best, which is consistent with the definition of the composite distribution. Both larger and smaller α values lead to suboptimal results, indicating that both exploration and exploitation play important roles in the CoVRL framework. We will add a more detailed discussion in the revised version.
>
> 1. **Degradation of 'Math Only':** This degradation is likely due to noise in the training data. Specifically, WebInstruct is derived through LLM-based extraction, and the extracted answers may not always be perfectly accurate. Consequently, training on the math-only subset may introduce noisier supervision, which could have a negative impact on certain benchmarks such as CARP-EN.
> 2. **Standard Deviation:** We conducted multiple runs on the small test set to ensure evaluation stability. We agree that reporting standard deviations would provide a clearer picture, and we will add these statistics in the revised version. For reference, the standard deviations of the main results are as follows:
>
> | Metrics/Dataset | GPQA       | MMLU-Pro  | TheoremQA | AIME’24 | AQuA | CARP-EN | MATH-500 | Minerva | SAT-Math |
> | --- | --- | --- | --- | --- | --- | --- | --- | --- | --- |
> | Acc | 0.304 | 0.465 | 0.363  | 0.075 | 0.773 | 0.651 | 0.663 | 0.255 | 0.971 |
> | Std | 0.021 | - | 0.004 | 0.027 | 0.010 | 0.007 | 0.011 | 0.020 | 0.038 |
> 1. **Name consistency:** Thank you for pointing out the typo. We will correct the naming inconsistency in the revised version.

---

> > ### Author Rebuttal · Reviewer_Wn7g · 2026-04-02
> >
> > A critical gap remains: the paper evaluates only final answers, not the correctness of the reasoning chains. In mathematical or deductive reasoning, a correct answer derived from flawed logic is misleading. Without process-level validation, the method’s true contribution to reasoning quality remains unsubstantiated.
> >
> > Thank you to the authors for your response!
> >
> > I will keep my score.

---

> > > ### Author Response · Authors · 2026-04-02
> > >
> > > Thank you for your suggestion. We agree that the quality of reasoning traces is important. Due to the inherent difficulty of evaluating reasoning processes and the time constraints of the rebuttal period, we were unable to involve human experts for evaluation. Instead, we adopt multiple open-source and closed-source models as LLM-as-Judge evaluators, scoring reasoning quality along four dimensions proposed by prior work ([Lee et al., 2025](https://arxiv.org/pdf/2502.12289)) on a scale of 1–5. The four dimensions are defined as follows:
> > >
> > > - **Factuality**: evaluates whether the factual information in a reasoning step can be grounded in reliable sources.
> > > - **Validity**: evaluates whether a reasoning step contains no logical errors.
> > > - **Coherence**: measures whether a reasoning step's preconditions are satisfied by the previous steps.
> > > - **Utility**: measures whether a reasoning step contributes to reaching the correct final answer.
> > >
> > > We randomly sampled 100 questions from MMLU-Pro for evaluation. Each LLM judge was queried 8 times per sample to report mean and standard deviation. The results are as follows:
> > >
> > > | Model/Evaluation | GPT-OSS factuality | GPT-OSS validity | GPT-OSS coherence | GPT-OSS utility | GPT-5.2 factuality | GPT-5.2 validity | GPT-5.2 coherence | GPT-5.2 utility | Claude-Sonnet-4.6 factuality | Claude-Sonnet-4.6 validity | Claude-Sonnet-4.6 coherence | Claude-Sonnet-4.6 utility |
> > > | --- | --- | --- | --- | --- | --- | --- | --- | --- | --- | --- | --- | --- |
> > > | Qwen2.5-7B-Base | 2.56 ± 0.04 | 2.82 ± 0.04 | 3.72 ± 0.03 | 2.68 ± 0.04 | 2.55 ± 0.03 | 2.73 ± 0.02 | 3.83 ± 0.02 | 2.90 ± 0.03 | 2.76 ± 0.04 | 2.69 ± 0.03 | 3.16 ± 0.03 | 2.48 ± 0.03 |
> > > | Qwen2.5-7B-Base after CoVRL | 2.79 ± 0.01 | 3.01 ± 0.04 | 3.77 ± 0.04 | 2.93 ± 0.04 | 2.72 ± 0.04 | 2.83 ± 0.02 | 3.87 ± 0.02 | 3.05 ± 0.02 | 3.01 ± 0.03 | 2.87 ± 0.02 | 3.26 ± 0.05 | 2.85 ± 0.03 |
> > >
> > > As shown above, after training with our method, the scores across all four dimensions improve consistently across all evaluator models. We believe this serves as evidence that our algorithm genuinely enhances reasoning quality. We will include more analysis in the revised version. Furthermore, our method shares an assumption similar to that of RLVR: if a model is able to derive the correct final answer through reasoning, its intermediate reasoning steps are generally sound. This is justified by the fact that the answer space is vast while correct answers are extremely sparse within it — making it highly unlikely that flawed reasoning consistently leads to correct answers by chance.

---

### Official Review · Reviewer_aKvh · 2026-03-14

**Soundness:** 3
**Presentation:** 3
**Significance:** 3
**Originality:** 3
**Overall Recommendation:** 4
**Confidence:** 4

**Summary:**

This paper tackles the challenge of "verifier-free" reinforcement learning for Large Language Models (LLMs). Existing methods sample reasoning traces conditioned only on the question (prior sampling) and use the likelihood of generating the correct answer as the reward, which suffers from low exploration efficiency and trace-answer incoherence. The authors propose CoVRL, which formulates reasoning optimization as a variational inference problem. It constructs a composite distribution that couples the prior (question-only) and posterior (answer-guided) distributions. To optimize this tractably, they employ a hybrid off-policy sampling strategy with importance weighting and a low-variance Bregman divergence-based KL estimator. Experiments on Qwen models (7B to 14B) demonstrate solid improvements over base models and existing verifier-free RL baselines on both mathematical and general reasoning tasks.

**Compliance With Llm Reviewing Policy:**

Affirmed.

**Key Questions For Authors:**

- Since you identify LLM-as-a-verifier as the primary alternative to your method, could you provide baseline results using an LLM-as-a-Judge (e.g., using the Qwen2.5-7B/14B model to score reasoning traces)? Does CoVRL actually suffer from less reward hacking than this baseline?

- Could you provide a quantitative comparison of the training time (e.g., GPU hours) and memory usage of CoVRL compared to standard GRPO and RLPR? Does the requirement for two forward passes per trace significantly bottleneck the RL pipeline?

**Strengths And Weaknesses:**

Strengths:

- The formulation of the reasoning generation process through the lens of variational inference is elegant. The derivations for the token-level importance sampling ratio and the low-variance KL estimators (using Bregman divergence as a control variate) are mathematically sound and well-adapted to the hybrid sampling strategy.

- In standard outcome-supervised RL (when only the final answer is known), exploring the massive space of possible reasoning chains without guidance is highly inefficient. Incorporating the posterior (answer-conditioned) generation to guide the trace generation, while mathematically adjusting for the distribution shift back to the prior, is a very clever solution to the trace-answer incoherence problem.

Weaknesses:

- The authors' claim of a 'verifier-free' framework is slightly misleading. Even though no external Reward Model is trained, the posterior distribution explicitly conditions on the ground-truth answer. Therefore, optimizing this posterior acts as an implicit form of reward modeling, merely replacing traditional evaluative critiques with answer-conditioned reference traces.

- Equation 12 and the surrounding text indicate that computing the importance ratio requires two forward passes for every sampled reasoning trace: one using the prior template and one using the posterior template. This effectively doubles the computational cost of the rollout.

- The authors motivate their verifier-free approach by claiming that using an LLM as a verifier "faces the risk of reward hacking" (Section 1). However, they do not include any LLM-as-a-verifier baselines in their experiments. Dismissing a prominent alternative approach purely on theoretical grounds without empirical comparison weakens the paper. To convincingly argue that CoVRL is a superior alternative, the authors must compare it against a standard LLM-as-a-Judge baseline (e.g., using the model itself)  to evaluate the actual trade-offs in performance, compute, and hacking properties.

---

> ### Author Rebuttal · Authors · 2026-03-31
>
> Thank you for acknowledging the novelty and soundness of our work.  We would like to address the reviewer’s concerns as follows.
>
> 1. **Verifier-Free claim:**
>
> As you noted, the KL term in Eq. 8 serves as a token-level process reward. Meanwhile, the reconstruction term in Eq. 8 acts as an outcome reward, playing a similar role to the outcome signal provided by a verifier in RLVR. However, unlike RLVR, which primarily targets mathematics and code tasks where robust rule-based verifiers are readily available, our method focuses on general reasoning tasks that typically lack such verifiers — even when reference answers are available. For example, a chemistry problem may require generating a reaction equation such as "2 CH₃COOH + CaC₃H₇O₆P → Ca(CH₃COO)₂ + C₃H₉O₆P", for which no reliable rule-based verification systems are currently available. By describing our framework as verifier-free, we mean that we directly leverage the model's probability of generating the correct answer as the outcome signal, thereby eliminating the need for robust rule-based verifiers. This allows the scope of reasoning training to be extended to a broad range of tasks.
>
> 1. **Computation Cost:**
>
> We first answer the question from a theoretical perspective. In our experiments, we keep the same batch size across methods. For baselines that sample only from the prior distribution, all trajectories are generated under the question-only setting. In contrast, for CoVRL, roughly half of the trajectories are sampled under the question-only setting and the other half under the answer-guided setting, in expectation. Assuming similar average response lengths, the rollout cost of CoVRL is therefore approximately comparable to that of standard GRPO. The additional computation mainly comes from two stages. First, during actor update, CoVRL optimizes both the prior and posterior distributions, which theoretically doubles the computation in this stage. Second, when computing old log-probabilities for PPO/GRPO-style updates, CoVRL also needs to evaluate both prior and posterior probabilities under the old policy, which introduces extra forward-pass cost. As for memory usage, modern RL libraries such as VeRL typically rely on efficient inference engines such as vLLM, so GPU memory usage during rollout remains essentially unchanged. During actor update, if we keep the micro-batch size fixed and correspondingly increase gradient accumulation steps, the peak memory usage can likewise remain largely unchanged.
>
> To quantify the overhead, we measure the time breakdown at step 1 using Qwen2.5-7B-Base + FSDP2 + vLLM, on 4 × 8 GPUs, with batch size = 192 and the maximum response length of 2048:
>
> | Setting/Stage | Generated batch | Compute reward | Compute advantage | Update actor | Compute old logprobs | Overall |
> | --- | --- | --- | --- | --- | --- | --- |
> | CoVRL | 21.94 sec. | 0.51 sec. | 0.03 sec. | 10.60 sec. | 3.01 sec. | 36.09 sec. |
> | GRPO | 21.70 sec. | 0.36 sec. | 0.03 sec. | 5.93 sec. | 2.30 sec. | 30.32 sec. |
>
> Consistent with the analysis, the actor-update cost is approximately doubled. However, since rollout still dominates the total step time, the end-to-end overhead is much smaller: the overall per-step time increases from 30.32 seconds to 36.09 seconds. We also note that under a fully on-policy implementation, old log-probabilities can be computed together with the actor-update forward pass. In that case, this stage would not introduce additional overhead specific to CoVRL relative to standard GRPO. We will add a computation cost discussion in the revised version.
>
> 1. **LLM-as-a-verifier:**
>
> We do not view the LLM-as-a-judge approach and CoVRL as strictly alternative choices. Instead, if we replace the target in Eq. 5 from $\log p_\theta(y \mid x)$ to $\log \underset{p(z,y \mid x)}{\mathbb{E}} e^{R(y)}$ (we introduce $\exp$ to handle negative rewards, and $R$ can be a reward function provided by an LLM judge), then Eq. 8 turns into $\underset{p'(z,y | x,y^{\ast})}{\mathbb{E}}[R(y)] - D_{\text{KL}}(p'(z,y | x,y^{\ast}) || p_\phi(z,y | x))$, where $y^*$ stands for the reference answer. Intuitively, the outcome reward is now provided by an LLM judge, while the KL loss offers fine-grained supervision for both the prior and posterior distributions. We supplement the experiment with an LLM-as-Judge verifier following the setting of the original [WebInstruct-verified](http://arxiv.org/abs/2505.14652), which employs a specifically trained judge model along with format penalty, length penalty, and KL reward:
>
> | Method/Performance | GPQA | MMLU-Pro | TheoremQA | AIME’24 | AQuA | CARP-EN | MATH-500 | Minerva | SAT-Math | Overall |
> | --- | --- | --- | --- | --- | --- | --- | --- | --- | --- | --- |
> | Base Model | 26.1 | 36.7 | 25.2 | 2.7 | 55.6 | 54.3 | 44.7 | 18.6 | 76.5 | 37.8 |
> | LLM-as-Judge | 30.5 | 42.4 | 27.4 | 3.0 | 59.6 | 55.9 | 49.4 | 20.1 | 84.6 | 41.4 |
> | CoVRL | 30.4 | 46.5 | 36.3 | 7.5 | 77.3 | 65.1 | 66.3 | 25.5 | 97.1 | 50.2 |

---

> > ### Author Rebuttal · Reviewer_aKvh · 2026-04-07
> >
> > Concerns resolved. I recommend clarifying the definition of “verifier-free” and the corresponding training setup more explicitly in the revised manuscript.

---

### Official Review · Reviewer_CHWG · 2026-03-21

**Soundness:** 2
**Presentation:** 2
**Significance:** 3
**Originality:** 3
**Overall Recommendation:** 4
**Confidence:** 3

**Summary:**

The authors tackle the challenge of improving LLM reasoning capabilities in domains where a verifier is unavailable. The paper identifies two major bottlenecks in the current verifier-free literature: (1) low sample efficiency, which requires generating large number of reasoning traces to find a correct logical path to the answer, and (2) potential incoherence, where the correct final answer may be expressed in a format different from the one naturally obtained via correct LLM reasoning, leading to inaccurate reward assignment.
To address these issues, the authors propose Coupled Variational Reinforcement Learning (CoVRL). The method applies a variational inference framework made tractable through specific heuristics. They introduce a composite distribution that couples the prior (reasoning conditioned purely on the question) and the posterior (reasoning conditioned on both the question and the correct answer). This is paired with a hybrid sampling strategy that samples from the prior and the posterior based on assigned probabilities. Finally, the authors demonstrate that CoVRL achieves a 12.4% performance improvement over the base model.

**Compliance With Llm Reviewing Policy:**

Affirmed.

**Key Questions For Authors:**

1) Justification for Composite Distribution Split: What is the intuition of 1/2 and 1/2​ weighting in the composite distribution (Equation 6)?
Answering this satisfactorily would strengthen the method's presentation.

2) Incoherence Claim vs. Experimental Setup: The motivation of the "potential incoherence" problem seems to require mathematical reasoning. However, lines 306-308 indicate  the experiments retain only the non mathematical questions. It is not clear why this justifies "potential incoherence" problem.
Clarifying this would strengthen the method's soundness.

3) Term "Verifier-Free": Since training requires the correct answer for posterior (which essentially acts as a verified outcome), could you explain which stages are truly "verifier-free"?
 This is one of my concern in the presentation.

4) Some Corrections: Figure 1 - Please address that the domain of sin^2(x)/(1+cos(x))​ is not the same as 1−cos(x) (the former is undefined at x=(2n+1)π) so the expressions are not mathematically equivalent as wrongly claimed to be.
                    Figure 2 - Typos "Thouhgt" and "Adavnatge" in Figure 2.
How this impacts my evaluation: This will not affect the final score, but fixing these seem necessary.

**Limitations:**

Yes

**Strengths And Weaknesses:**

Soudness: The submission is technically sound, with claims generally well-supported by experimental results and a clear mathematical formulation. However, there seems to be an empirical disconnect regarding the "potential incoherence" problem. While the paper motivates this as an issue primarily found in mathematical reasoning, lines 306-308 indicate the experiments only retain non-mathematical questions. So there seems to be a disconnect between the motivation and this part.

Presentation: The submission is well-written and the overall narrative is easy to follow, but perhaps clarity could be improved for the following:
a) The decision to use a 1/2 and 1/2 split for the composite distribution does not seem to be well-argued in the text.
b) The hybrid sampling and KL estimators seems to lack sufficient discussion. The authors could perhaps dedicate more space to explaining these in the main paper.
c) The term "verifier-free" is somewhat unclear. Since the training requires the correct answer (which may act as a verified outcome), it would be beneficial if more discussion on which stages are truly verifier-free is provided.
d) There are some typos such as, "Thouhgt" and "Adavnatge" in Figure 2.

Significance: The paper does address a relevant problem of LLM reasoning. In my opinion it advances understanding and practice in machine learning, and could influence future research. While the scope is specialized to reasoning tasks, this is appropriate for the contribution, and the empirical improvements demonstrate clear practical utility.

Originality: The work provides new insights with the new approach CoVRL. The work does offer novel contribution, and the novelty is also justified.

---

> ### Author Rebuttal · Authors · 2026-03-31
>
> Thank you for acknowledging the novelty of our work. We would like to address the reviewer’s concerns as follows.
>
> 1. **Composite Distribution Setting:**
>
> We use a 1/2–1/2 split for the composite distribution as a symmetric coupling between the prior and posterior. The two distributions play complementary roles: the prior reflects inference-time generation, while the posterior provides answer-guided exploration during training. An equal weighting therefore provides a neutral balance between these two objectives. This design is also motivated by the Jensen–Shannon divergence (JSD), which can be viewed as a symmetrized and smoothed variant of the KL divergence. Specifically, JSD introduces the mixture distribution $M=\frac{1}{2}(P+Q)$, and is defined as $\mathrm{JSD}(P|Q)=\frac{1}{2}\mathrm{KL}(P|M)+\frac{1}{2}\mathrm{KL}(Q|M)$. In CoVRL, our composite distribution plays a similar role as a coupled intermediate distribution between the prior and posterior, allowing both components to participate in optimization and receive gradient updates. However, our ultimate goal is to improve the prior distribution, i.e., to answer the question correctly without answer guidance. For this reason, the regularization term in Eq. 8 cannot be directly replaced by JSD.  However, if one replaces $p'(z|x,y)$ with $p_{\text{hybrid}}(z|x,y)$ under $\alpha=0.5$, and replaces $p_\phi(z|x)$ with $p'(z|x,y)$, then the resulting KL objective will become the JSD form. Under that formulation, the optimization target would become the composite distribution $p'(z|x,y)$ rather than the prior distribution $p_\phi(z|x)$, while the variational distribution would be the hybrid distribution $p_{\text{hybrid}}(z|x,y)$. Such a formulation may be useful in settings where the performance of both the prior and posterior distributions is equally important, and we believe it is an interesting direction for future investigation. We will add the relevant derivation and discussion in the revised version.
>
> 2. **The hybrid sampling and KL estimators:**
>
> Thank you for the suggestion. In the current version, the derivation of the KL divergence estimator under the hybrid sampling strategy is provided in Appendix A. We will add further explanation to the main text in the revised version.
>
> 3. **Incoherence Claim vs. Experimental Setup:**
>
> Trace–answer incoherence refers to cases where the model’s reasoning trace leads to an answer that is semantically equivalent to the reference answer but differs in surface form. As the model performs next-token prediction conditioned on its own reasoning trace, this mismatch can cause it to assign a low probability to the reference answer. This issue is not limited to mathematical reasoning, but can also arise in general reasoning tasks. For example, in a physics problem, the reference answer may be “0.5 kg”, while the model’s reasoning trace may naturally yield “500 g”. Although semantically equivalent, the difference in form may still lower the probability assigned to the reference answer. We focus on non-mathematical data primarily to evaluate the effectiveness of our verifier-free method in domains where no robust verifier is available. We discuss this point in more detail below.
>
> 4. **Term "Verifier-Free":**
>
> We use the term verifier-free primarily to distinguish our method from prior RLVR-based approaches. Existing RLVR methods are typically applied to domains such as mathematics and code generation, where a highly reliable rule-based verifier can be used to assess the correctness of an output. For instance, rule libraries such as Math-Verify can be used to check answer equivalence under predefined rules. In code generation, correctness can be evaluated through sandboxed execution against test cases. The availability of such verifiers facilitates reinforcement learning training and helps avoid the risk of reward hacking. However, for broader reasoning tasks, including physics and chemistry, even relatively short-form answers may not admit a simple verifier with sufficiently high accuracy. For example, a chemistry problem may require generating a reaction equation such as “2 CH₃COOH + CaC₃H₇O₆P → Ca(CH₃COO)₂ + C₃H₉O₆P”, for which there may be no readily available rule-based system capable of reliably verifying correctness. To address this challenge, our method avoids reliance on a pre-specified external verifier by treating the model’s probability of generating the correct answer as the reward signal under a variational inference formulation. Although this signal is conditioned on the target answer, it does not rely on a separate robust verification module, and is therefore less susceptible to the reward-hacking issues that may arise from imperfect learned verifiers. In this sense, our method is *verifier-free*.
>
> 5. **Typos:**
>
> Thank you for pointing this out. We will replace the example in Figure 1 with a more appropriate one, and we will also correct the typos in Figure 2 in the revised version.

---

> > ### Author Rebuttal · Reviewer_CHWG · 2026-04-04
> >
> > Thanks for the rebuttal. I would like to keep my score of 4 after reading the other reviews.

---

### Decision · Program_Chairs · 2026-04-30

**Decision:**

Accept (regular)

**Comment:**

This paper received generally positive feedback from the reviewers, who highlighted several key strengths:

* The paper tackles an important and practical problem—improving reasoning in settings without explicit verifiers—by proposing a principled variational formulation that is both novel and well-motivated.
* The CoVRL framework introduces an elegant coupling of prior and posterior reasoning distributions, offering a thoughtful solution to both exploration inefficiency and reasoning–answer incoherence.
* Empirical results are solid across multiple reasoning benchmarks and model scales, demonstrating consistent improvements over existing verifier-free RL baselines.

At the same time, reviewers noted a few areas for improvement. The notion of “verifier-free” could be clarified more precisely given the use of answer-conditioned signals, and additional comparisons to stronger baselines (e.g., verifier-based or LLM-as-judge approaches) would strengthen the empirical claims. Some aspects of the evaluation—such as reasoning quality analysis, parameter sensitivity, and computational cost—would also benefit from deeper investigation to improve confidence in robustness and generality.

During the rebuttal, the authors addressed several concerns around clarity, experimental setup, and positioning, which helped improve understanding of the method and resolve some reviewer questions, particularly regarding the training formulation and empirical comparisons. However, some issues remain only partially addressed, including the precise definition of “verifier-free,” the breadth of baseline comparisons, and deeper analysis of reasoning correctness and efficiency trade-offs.

Overall, the paper presents a novel and meaningful contribution with promising empirical results, though some aspects could be further strengthened. Therefore, I recommend weak acceptance.